# Defending Against Unknown Corrupted Agents: Reinforcement Learning of Adversarially Robust Nash Equilibria

**Andi Nika**                                                                                      *andinika@mpi-sws.org*
*Max Planck Institute for Software Systems*

**Jonathan Nöther**                                                          *s8jonoet@stud.uni-saarland.de*
*University of Saarland & Max Planck Institute for Software Systems*

**Adish Singla**                                                                                   *adishs@mpi-sws.org*
*Max Planck Institute for Software Systems*

**Goran Radanović**                                                                      *gradanovic@mpi-sws.org*
*Max Planck Institute for Software Systems*

**Reviewed on OpenReview:** *https://openreview.net/forum?id=aggyMifxLQ*

## Abstract

We consider a Multi-agent Reinforcement Learning (MARL) setting, in which an attacker can arbitrarily corrupt any subset of up to $k$ out of $n$ agents at deployment. Our goal is to design agents that are robust against such an attack, by accounting for the presence of corrupted agents at test time. To that end, we introduce a novel solution concept, the Adversarially Robust Nash Equilibrium (ARNEQ), and provide theoretical proof of its existence in general-sum Markov games. Furthermore, we introduce a proof-of-concept model-based approach to computing it and theoretically prove its convergence under standard assumptions. We also present a practical approach called Adversarially Robust Training (ART), an independent learning algorithm based on stochastic gradient descent ascent. Our experiments in both cooperative and mixed cooperative-competitive environments demonstrate ART's effectiveness and practical value in enhancing MARL resilience against adversarial behavior.

## 1 Introduction

The growing prevalence of automated systems has facilitated a fertile ground for implementing the celebrated multi-agent reinforcement learning (MARL) framework to solve a wide range of important problems. Prominent examples include finance (Shavandi & Khedmati, 2022; Lee et al., 2007), sensor networks (Cortes et al., 2004; Choi et al., 2009), autonomous vehicles (Shalev-Shwartz et al., 2016; Zhou et al., 2020; Palanisamy, 2020) and gaming (Vinyals et al., 2019; Perolat et al., 2022).

One of the main underlying assumptions of this practical framework has been the inherently rational and selfish nature of the learning agents, according to which their sole objective is to optimize their utilities, an assumption stemming from game theory (Osborne & Rubinstein, 1994) and economics since the time of Smith (1776). Indeed, the overwhelming majority of MARL successes have been made by algorithms that are designed to optimize their objectives.

However, this assumption has been challenged by different lines of research, only sporadically in the past, but increasingly more in recent years. For example, Chen et al. (2023) study the problem of robustness of distributed RL systems against *Byzantine attacks*, where a fraction of agents can report fake data to

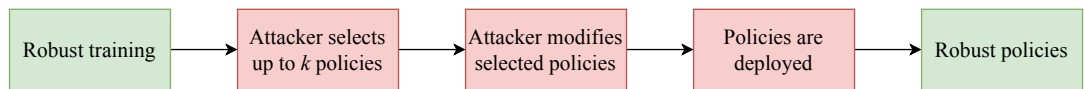

Figure 1: Attack-defense interaction is depicted. First, the agents are trained without any adversarial threat. Then, the attacker observes all the trained policies of the agents and arbitrarily picks any up to $k$ of them. Once the chosen policies are arbitrarily perturbed, they are deployed. Our goal is to train policies that remain robust under attack.

the center. Another line of research, which has gained a lot of popularity recently, studies the problem of *adversarial policies* in MARL (Guo et al., 2021), where an agent faces an adversarial opponent trained in a zero-sum Markov game regime. Eliaz (2002) considers the problem of implementing games with potentially *faulty agents*, that is, agents who do not necessarily aim at optimizing their utilities but might behave erratically due to some system malfunction. Furthermore, Babaioff et al. (2007) consider congestion games with *malicious players* and study the game-theoretic properties of such games.

The shared underlying insight among these various lines of work is that in reality, certain agents could exhibit unpredictable behavior, stemming from system malfunctions or unanticipated adversarial objectives. This insight is further evidenced in various safety-critical applications of MARL, where the presence of such agents might seriously deteriorate their performance. For example, in traffic management (Kuyer et al., 2008) and autonomous driving systems, some agents, be it autonomous or human, might cause traffic jams or fatal accidents due to their erratic behavior, while in distributed energy systems (Roesch et al., 2020) they might compromise the stability and reliability of the power grid.

Be that as it may, the aforementioned previous work lacks a simultaneously practical and conceptually general treatment of the problem of malicious agents in multi-agent systems. While Eliaz (2002) proposes an interesting solution concept, that of a faulty-tolerant Nash equilibrium, their focus is on the implementation problem, not a learning one, so the question of efficiently computing such an equilibrium is left open. The line of work initiated by Gleave et al. (2020) on adversarial policies focuses on two-player games, and their considerations are mostly from the attack's perspective. Chen et al. (2023) consider distributed RL systems, where agents share the same goal, while Babaioff et al. (2007) study the properties of a specific class of games with malicious players, with fixed identities.

Motivated by the above shortcomings, we are interested in providing a formulation of the problem that captures a general attack model. In particular, we are interested in a model that: (i) involves $n$-player Markov games; (ii) the attacker can manipulate any up to $k$ out of $n$ trained policies; (iii) the attacker can arbitrarily modify the controlled policies. Our aim is to find an efficient defense that takes into account such an attack model and serves as a robust training procedure for the learning agents.

To that end, we formulate a general-sum Markov game with $n$ players, whereby an attacker is given access to all agents' policies at test-time, previously trained on a clean environment. The attacker then arbitrarily picks and perturbs *any up to $k < n$* of them, at which point all policies are deployed. Ultimately, we are interested in devising a robust training procedure that accounts for the potential attack at test time and can serve as a defense against it. The attack-defense model is described in Figure 1. Below we describe our main contributions.

- First, in the same spirit as (Eliaz, 2002; Zhang et al., 2020), we introduce a new notion of equilibrium, the $k$-Adversarially Robust Nash Equilibrium (ARNEQ), designed to be robust against the worst-case arbitrary perturbations of *any $k$* out of $n$ policies, and show its existence in general-sum Markov games.
- Second, we propose a model-based procedure to compute such a strategy and show that it converges to an ARNEQ under standard assumptions on the game structure.
- Third, motivated by gradient-based decentralized methods in Markov games, we propose Adversarially Robust Training (ART) for MARL, an efficient gradient-based learning algorithm, that can be run independently among all players at training time.
- Finally, we provide extensive experiments on both cooperative and mixed environments that showcase the efficiency of ART as a robust defense. Our results show that ART converges to a stable joint strategy and is able to substantially improve performance, which can be clearly observed in our qualitative results.

### 1.1 Other Related Work

#### 1.1.1 Attacks and defenses in (MA)RL.

Adversarial attacks on RL systems have been extensively studied in recent years (Kiourti et al., 2020; Lin et al., 2017; Huang et al., 2017; Mohammadi et al., 2023; Sun et al., 2020a;b). The main types of these attacks include training-time attacks (Rakhsha et al., 2020; Xu et al., 2021), test-time attacks (Behzadan & Munir, 2017; Huang et al., 2017; Kos & Song, 2017; Sun et al., 2020a), and backdoor attacks (Kiourti et al., 2020; Wang et al., 2021; Yang et al., 2019), where the adversarial perturbations happen at the environment level. More recently, however, there has been a growing interest in the so-called adversarial policies in (MA)RL (Gleave et al., 2020; Guo et al., 2021; Liu et al., 2022; Wang et al., 2023; Mohammadi et al., 2023; Li et al., 2019) where attackers assume the identity of an agent, and thus attack by inducing natural observations. Our work belongs to this class of attack type, but generalizes the framework of adversarial policies in MARL for general $n$-player games and focuses on the defense front. Various types of defenses have been proposed against test-time attacks (Zhang et al., 2020; Pattanaik et al., 2017; Zhang et al., 2021a; Wu et al., 2021), training-time attacks (Banihashem et al., 2023; Kumar et al., 2021; Lykouris et al., 2021; Wu et al., 2022; Nika et al., 2023; Zhang et al., 2021a; 2022) and backdoor attacks (Bharti et al., 2022). When it comes to adversarial policies, previous work has mainly focused on heuristic defenses against them. For two-player zero-sum Markov games, Gleave et al. (2020) propose sequential fine-tuning procedures against various adversaries, which seems to robustify the trained policies against such adversaries, but not against a more general attack. Furthermore, apart from issues such as catastrophic forgetting and myopic robustness against a specific class of adversaries, the aforementioned defense strategy becomes combinatorially expensive in $n$-player Markov games. If the identities of the controlled agents and the objectives of the attacker were known, then a natural solution concept would be the Nash equilibrium (NE) strategy of the new game. In our setting, this information is not given. As a consequence, the problem is no longer a standard Markov game, and thus there is no NE defined in its solution space. Recently Liu et al. (2023) consider the problem of defense in the two-player regime with one adversarial policy and take a game-theoretic approach. They define a new game with modified utilities and then solve the game as if it were a zero-sum game. We generalize their setting and study the defense problem against any up to $k$ out of $n$ adversarial policies in MARL. As we mentioned earlier, this formulation renders the classical NE notion inappropriate. Thus, a new solution concept is necessary.

#### 1.1.2 Robust (MA)RL.

Another line of research related to our work is that of robustness in (MA)RL under model uncertainty (Wang & Zou, 2021; Zhang et al., 2020; Russel et al., 2020; Mankowitz et al., 2018). In this line of work, the focus is on designing algorithms that learn under worst-case environment assumptions. Instead, we consider the robustness against the worst-case subset of adversarial policies at test time. A recent work from Li et al. (2023) studies a similar problem to ours. They consider defenses against unknown Byzantine attacks in a cooperative regime. However, they take a different approach to the defense problem. They capture the uncertainty about the adversaries as uncertainty in transitions, which are characterized by types, thus taking a Bayesian game approach, while allowing only for one adversary. We take a direct robust approach on each individual agent while allowing for a more general attack of up to $k$ adversaries. Furthermore, their notion of robust equilibrium pertains to only one joint adversarial policy, while our robust equilibrium associates each robust policy with a specific joint adversarial strategy. This substantially changes the setting since we are, in effect, solving $n$ games simultaneously.

#### 1.1.3 Markov games.

We formalize our study using the notion of Markov games (Shapley, 1953). Our goal is to design decentralized robust training procedures. There has been a lot of research in the past decade on model-based centralized (Hu & Wellman, 2003) and model-free decentralized methods of computing the celebrated Nash equilibrium of such games (Daskalakis et al., 2020; Leonardos et al., 2021; Ding et al., 2022; Zeng et al., 2022; Giannou et al., 2022; Zhang et al., 2021b; Wang & Zou, 2022; Wei et al., 2021). However, all efficient methods are proposed for particular types of Markov games, such as zero-sum or potential Markov games. The notion of

robust Markov games has also been previously considered. For instance, Aghassi & Bertsimas (2006) propose the notion of ex-post equilibrium to handle payoff uncertainty in normal form games, Kardeş et al. (2011) consider robust Markov games with model uncertainty. In contrast, our notion of uncertainty is related to adversarial policies. Finally, Kalogiannis et al. (2023) consider cooperative Markov games with a fixed adversary. Our work differs from them in that we consider a general-sum Markov game setting and assume the presence of unknown adversarial policies. This substantially changes the nature of the game since the agents cannot cooperatively team up against the adversaries.

## 2 Problem Formulation

In this section, we first introduce the relevant notation, then give the necessary background of the problem, after which we formally introduce our problem.

**Notation.** As usual $\langle \cdot, \cdot \rangle$ will denote the inner product between two vectors from the same vector space, $[n]$ denotes the set of natural numbers up to and including $n$. We denote by $\|\cdot\|$ the Euclidean norm and by $\Delta(X)$ the set of probability simplices on the set $X$.

### 2.1 Preliminaries

Let $\mathcal{G} = (\mathcal{N}, \mathcal{S}, \mathcal{A}_1 \times \ldots \times \mathcal{A}_n, \mathcal{R}_1, \ldots, \mathcal{R}_n, \mathcal{P}, \gamma, \mu)$ be a Markov game. Here $\mathcal{N} = [n]$ represents the set of $n$ players, $\mathcal{S}$ denotes the state space with cardinality $S$, $\mathcal{A} := \mathcal{A}_1 \times \ldots \mathcal{A}_n$ is the joint action space with $\mathcal{A}_i$ denoting the action space of player $i$ with cardinality $A_i$. The reward of player $i$ is a function of the state and joint action, i.e., $\mathcal{R}_i : \mathcal{S} \times \mathcal{A} \to [0, 1]$. The transition kernel is given by $\mathcal{P} : \mathcal{S} \times \mathcal{A} \to \Delta(\mathcal{S})$, with $\mathcal{P}(s'|s, a)$ denoting the probability that the game transitions into state $s'$ given that joint action $a$ is taken in state $s$. The discount factor is denoted by $\gamma \in [0, 1)$, and $\mu \in \Delta(\mathcal{S})$ denotes the initial state distribution. Finally, we will denote by $\mathcal{N}_i$ the set of subsets of cardinality no greater than $k$ of $\mathcal{N} \setminus \{i\}$, and we let $\Omega_i = \Delta(\mathcal{N}_i)$.

Stationary policies are mappings from states to distributions over actions. Formally, we let $\pi_i : \mathcal{S} \to \Delta(\mathcal{A}_i)$ be a stationary policy for player $i$, lying in the policy space $\Pi_i$. Moreover, we denote by $\Pi = \Pi_i \times \ldots \times \Pi_n$ the joint policy space of all players, with elements denoted by $\pi$. In Sections 4 and 5, we will consider neural policy classes. Given action $a$, policy $\pi$ and a set $K \in \mathcal{N}$, we define $a_K = (a_j)_{j \in K}$ and $a_{-K} = (a_j)_{j \notin K}$ for action $a$ and $\pi_K = (\pi_j)_{j \in K}$ and $\pi_{-K} = (\pi_j)_{j \notin K}$ for policy $\pi$.

The value function represents the expected discounted cumulative reward of a given player with respect to a given joint policy, starting from a given state. Formally, given policy $\pi \in \Pi$ and state $s \in \mathcal{S}$, the value function with respect to player $i$ is given by $V_i^s(\pi) = \mathbb{E}\left[\sum_{t=0}^{\infty} \gamma^t \mathcal{R}_i(s_t, a_t) \Big| s_0 = s, \pi, \mathcal{P}\right]$, where the sequence $s_0, a_0, s_1, a_1, \ldots$ denotes the traversed state-action tuples when the initial state is $s$, actions are taken using $\pi$ and the transitions follow $\mathcal{P}$. Furthermore, we let $V_i^\mu(\pi) = \mathbb{E}_{s \sim \mu}[V_i^s(\pi)]$ denote the value function with respect to the initial state distribution. The value function satisfies the Bellman equation:

$$V_i^\mu(\pi) = \sum_{s \in \mathcal{S}} \mu(s) \sum_{a \in \mathcal{A}} \prod_{j \in \mathcal{N}} \pi_j(a_j|s)(\mathcal{B}V_i)(s, a) , \tag{1}$$

with $\mathcal{B}$ representing the Bellman operator acting on $V_i$, defined as $(\mathcal{B}V_i)(s, a) := \mathcal{R}_i(s, a) + \gamma \langle \mathcal{P}(s, a), V_i(\pi) \rangle$, where $V_i(\pi)$ denotes the $S$-dimensional vector with values $V_i^s(\pi)$ and $\mathcal{P}(s, a)$ denotes the $S$-dimensional vector with entries $\mathcal{P}(s'|s, a)$, for all $s' \in \mathcal{S}$. Next, we formally define the notion of a Nash equilibrium.

**Definition 1** *A joint policy $\pi^*$ is said to be a Nash equilibrium (NE) strategy if no player can be better off by deviating from it. Formally, we have $V_i^\mu(\pi^*) \geq V_i^\mu(\pi_i', \pi_{-i}^*)$, for all $\pi_i' \in \Pi_i$ and $i \in \mathcal{N}$. Here $\pi_{-i}^*$ denotes the joint policy of the players other than $i$.*

*Further, a joint strategy $\pi^*$ is said to be an $\epsilon$-approximate NE strategy if $V_i^\mu(\pi^*) \geq V_i^\mu(\pi_i', \pi_{-i}^*) - \epsilon$, for all $\pi_i' \in \Pi_i$ and $i \in \mathcal{N}$. If an NE joint strategy is stochastic, it is said to be a mixed NE strategy.*

Fink (1964) showed that a mixed NE strategy always exists in any given $n$-player Markov game.

## 2.2 Attack Model

As already described in Section 1, we consider test-time attacks, whereby an attacker is allowed access to all $n$ trained policies and can arbitrarily pick and manipulate any up to $k < n$ of them. It's important to highlight that the attacker's ability to select any subset of $k$ trained policies adds an additional layer of complexity to the defense problem. Any information about the attack is necessary to obtain an efficient defense against it. In our setting, the learning agents are unaware of who among them will be attacked at test time, and so, do not know a priori which agents to treat as adversarial.

## 2.3 Defense Objective

Our focus in this paper is on the defense front. In particular, we are interested in devising training procedures that take into account the potential future threat at deployment. At training time, there is no attack present. The only information the agents have about the attack is the parameter $k$ and its realization at test time. Thus, our aim is to propose a training procedure for each individual agent, taking into account this parameter, and the fact that any other agent may behave adversarially towards it. In the next section, we will introduce a new solution concept that is designed to capture the nature of the attack and provide such a robust defense.

## 3 Theoretical Results

In this section, we first formulate our solution concept, namely, the adversarially robust Nash equilibrium. Then, we show its existence in general-sum Markov games. Finally, we provide a model-based approach to its computation and prove its convergence.

## 3.1 The Adversarially Robust Nash Equilibrium

In this section we introduce our solution concept which is tailored to our attack model and defense objective. Our goal here is to provide the agents with a defense strategy that they can use *independently* of the other agents. Similar to robust MARL with model uncertainty (Zhang et al., 2020), we consider robustness in MARL under *attack uncertainty*. The best that player $i$ can do with the given information is to solve the worst-case problem, i.e. to be robust against the worst-case attacker (in terms of the agents under attack and the objectives). To that end, we propose the Adversarially Robust Nash Equilibrium (ARNEQ), defined as follows.

**Definition 2** *A joint policy $\overline{\pi}^*$ is said to be an ARNEQ if there exist $S$-dimensional vectors $V_1^*, \ldots, V_n^*$ such that, for every player $i \in \mathcal{N}$ and state $s \in \mathcal{S}$, we have*

$$\overline{\pi}_i^*(\cdot|s) \in \arg\max_{\pi_i \in \Pi_i} \min_{\omega_i \in \Omega_i} \min_{\widehat{\pi}^i \in \Pi_{-i}} \mathbb{E}_{K \sim \omega_i} \left[ \sum_{a \in \mathcal{A}} \pi_i(a_i|s) \prod_{j \in K} \widehat{\pi}_j^i(a_j|s) \prod_{l \notin K \cup \{i\}} \overline{\pi}_l^*(a_l|s)(\mathcal{B}V_i^*)(s, a) \right],$$

*where we denote by $\widehat{\pi}^i$ the joint adversarial policy $(\widehat{\pi}_j^i)_{j \neq i}$ with respect to player $i$, where $\widehat{\pi}_j^i \in \Pi_j$, for each $j \neq i$, and $\Omega_i = \Delta(\mathcal{N}_i)$.*

Before we proceed any further, it is important to point out that this notion of equilibrium does not correspond to that in zero-sum Markov games, or even adversarial team Markov games. In fact, the game induced by the defensive behavior of the agents is not even a static game which all agents share with each other. To see this, consider the following example.

**Example 1** *Let $n = 3$ and $k = 1$. When computing an ARNEQ, each agent has to consider its worst-case adversary. Suppose that the worst-case adversary of agent 1 may be agent 2, while the worst-case adversary of agent 2 may be agent 3. So agent 1 and agent 2 are not playing the same game, since the benign agents and worst-case adversarial agents for each of them are different.*

We will first show that such an equilibrium always exists in general-sum Markov games. We defer the proof of the result to the Appendix (see the Supplementary Material).

**Theorem 1** *Let $\mathcal{G}$ be a finite general-sum Markov game with $n$ players, that is, suppose $\max\{n, S, A\} < \infty$, where $A = \max_{i \in [n]} A_i$, and let $k < n$. Assume that an arbitrary subset of at most $k$ players use arbitrarily modified utilities, and assume that the mixed strategies of all players lie in compact sets. Then, an ARNEQ, as in Definition 2, exists.*

The next natural question is how to find such an equilibrium. In the next section, we will focus on devising a model-based centralized approach to computing an ARNEQ and prove its convergence.

### 3.2 Adversarially Robust Nash Q-Learning

Nash Q-Learning (Hu & Wellman, 2003) is a model-based approach to finding the Nash equilibrium of a Markov game. Essentially, each player maintains estimates of the Q-values of every player[1] and then computes the Nash equilibrium of the stage game. The Q-values are then updated based on the outcome of applying the so-called Nash operator (analogous to the Bellman operator for Markov games) on the current Q-values. This method leverages a celebrated result from Filar & Vrieze (2012) which links the value functions that correspond to the Nash equilibrium of the entire Markov game with those corresponding to each individual stage game, which will be defined later.

First, we need to define an operator that is appropriate for our setting and satisfies the recursive property analogous to the Bellman operator. Based on Definition 2, we define the following update sequence, for any $i \in \mathcal{N}$, state $s \in \mathcal{S}$ and $t \geq 0$:

$$\overline{V}_i^{t+1}(s) = \max_{\pi_i} \min_{\substack{\omega_i \in \Omega_i \\ \widehat{\pi}^i \in \Pi_{-i}}} \mathbb{E}_{K \sim \omega_i} \left[ \sum_{a \in \mathcal{A}} \pi_i(a_i|s) \overline{\pi}_{-(K \cup \{i\})}^t (a_{-(K \cup \{i\})}|s) \widehat{\pi}_K^i(a_K|s)(\mathcal{B}\overline{V}_i^t)(s,a) \right] ,$$

where $\overline{\pi}^t$ denotes the joint benign policy computed at step $t$. Similarly, we define the Bellman backup for the Q-values with respect to an ARNEQ policy $\overline{\pi}^*$, for any state-action tuple $(s, a)$, as follows, for given learning rates $\alpha_t \in [0, 1)$:

$$\overline{Q}_i^*(s,a) = \mathcal{R}_i(s,a) + \gamma \sum_{s' \in \mathcal{S}} \mathcal{P}(s'|s,a) \mathbb{E}_{K \sim \omega_i^*} \left[ \sum_{a' \in \mathcal{A}} \overline{\pi}_{-K}^*(a'_{-K}|s') \widehat{\pi}_K^{i,*}(a'_K|s') \overline{Q}_i^*(s',a) \right] ,$$

where $\omega_i^*$ and $\widehat{\pi}^{i,*}$ are the adversarial counterparts of the ARNEQ policy of player $i$, and $\widehat{\pi}_K^{i,*} = (\widehat{\pi}_j^{i,*})_{j \in K}$. At this point, we need to introduce another relevant definition – an instantiation of the ARNEQ for a stage game defined in terms of the Q-values only for a given state. Our procedure will iteratively compute it for each encountered state.

**Definition 3** *Fix state $s \in \mathcal{S}$. Given the Q-values $\overline{Q}_i(s, a)$, for all $i \in \mathcal{N}$ and $a \in \mathcal{A}$, the joint tuple $(\overline{\pi}_i, \widehat{\pi}^i, \omega_i)_{i \in \mathcal{N}}$ is said to be a stage ARNEQ with respect to state $s$ if, for every $i \in \mathcal{N}$, we have*

$$(\overline{\pi}_i, \widehat{\pi}^i, \omega_i) \in \arg \max_{\pi_i \in \Pi_i} \min_{\substack{\omega'_i \in \Omega_i \\ \pi'_{-i} \in \Pi_{-i}}} \mathbb{E}_{K \sim \omega'_i} \left[ \sum_{a \in \mathcal{A}} \pi_i(a_i|s) \overline{\pi}_{-(K \cup \{i\})}(a_{-(K \cup \{i\})}|s) \pi'_K(a_K|s) \overline{Q}_i(s,a) \right] .$$

Now we are ready to formulate the update rule for our procedure as follows:

$$\overline{Q}_i^{t+1}(s_t, a_t) = (1 - \alpha_t) \overline{Q}_i^t(s_t, a_t) + \alpha_t \left( \mathcal{R}_i(s_t, a_t) + \gamma ARNEQ \left( \overline{Q}_i^t(s_{t+1}) \right) \right) , \tag{2}$$

where

$$ARNEQ(\overline{Q}_i^t(s_{t+1})) := \mathbb{E}_{K \sim \omega_i^t} \left[ \sum_{a' \in \mathcal{A}} \overline{\pi}_{-K}^t(a'_{-K}|s_t) \widehat{\pi}_K^{i,t}(a'_K|s_t) \overline{Q}_i^t(s_{t+1}, a') \right] ,$$

---

[1]We will use player and agent interchangeably.

with $\overline{Q}_i^t(s_{t+1}) = [\overline{Q}_i^t(s_{t+1}, a)]_{a \in \mathcal{A}}$. Here, $a_t$ denotes the joint action taken at time step $t$ in state $s_t$ by all the agents, $\overline{\pi}^t$ denotes the stage game ARNEQ joint policy computed with values $(\overline{Q}_i^t)_{i \in \mathcal{N}}$ with adversarial counterparts $(\omega_i^t, \widehat{\pi}^{i,t})_{i \in \mathcal{N}}$.

Before stating the main result of this section, we first state the assumptions under which it holds. They are standard in similar model-based approaches to equilibrium computation (Zhang et al., 2020; Hu & Wellman, 2003; Yang et al., 2018).

**Assumption 1** *Each state and action tuple has been visited infinitely often.*

**Assumption 2** *The rates $\alpha_t$ satisfy, for every $t \geq 0$:*

- $0 \leq \alpha_t < 1$, $\sum_{t \geq t} \alpha_t = \infty$ and $\sum_{t \geq 0} \alpha_t^2 < \infty$.

- $\alpha_t = 0$ for any $(s, a) \neq (s_t, a_t)$, that is, we only update the Q-values corresponding to traversed state-actions.

**Assumption 3** *For each stage game, one of the following conditions holds:*

- *A stage ARNEQ is also a global optimum, that is, for any $i \in \mathcal{N}$ and $\pi(\cdot|s)$, we have*

$$\mathbb{E}_{K \sim \omega_i^t} \left[ \sum_{a \in \mathcal{A}} \pi_{-K}^t(a_{-K}|s) \widehat{\pi}_K^{i,t}(a_K|s) \overline{Q}_i^t(s, a) \right] \geq \mathbb{E}_{K \sim \omega_i} \left[ \sum_{a \in \mathcal{A}} \pi_{-K}(a_{-K}|s) \widehat{\pi}_K^i(a_K|s) \overline{Q}_i^t(s, a) \right] ,$$

*for any $\pi \in \Pi$, $\widehat{\pi}^i \in \Pi_{-i}$ and $\omega_i \in \Omega_i$.*

- *A given player's payoff is increased if other benign players or its attacker deviate, that is, for any $i \in \mathcal{N}$ we have*

$$\mathbb{E}_{K \sim \omega_i^t} \left[ \sum_{a \in \mathcal{A}} \pi_{-K}^t(a_{-K}|s) \widehat{\pi}_K^{i,t}(a_K|s) \overline{Q}_i^t(s, a) \right] \leq \mathbb{E}_{K \sim \omega_i'} \left[ \sum_{a \in \mathcal{A}} \pi_i^t(a_i|s) \pi_{-(K \cup \{i\})}'(a_{-(K \cup \{i\})}|s) \widehat{\pi}_K'(a_K|s) \overline{Q}_i^t(s, a) \right] ,$$

*for all $\pi' \in \Pi$, $\widehat{\pi}' \in \Pi_{-i}$ and $\omega_i' \in \Omega_i$.*

Now we are ready to state the result. Its full proof is deferred to the Appendix (see the Supplementary Material).

**Theorem 2** *Under Assumptions 1, 2 and 3 (formally stated in the Appendix), the Adversarially Robust Nash Q-Learning procedure as given in Equation equation 2 converges to $\overline{Q}_i^*$, for every player $i \in \mathcal{N}$.*

*Proof sketch.* The main ingredient of the proof is utilizing a previous result from (Hu & Wellman, 2003) which states that, if Assumptions 1 and 2 hold, and a given operator on the Q-functions is a contraction, then the procedure described above converges to an equilibrium. So the only thing to prove is that the ARNEQ operator is a contraction. We use Assumption 3 to that end, by separately considering both cases of the assumption. With this, all the conditions of the utilized result are satisfied, and thus we conclude convergence to an ARNEQ. □

This result completes the theoretical characterization of our problem. Although the proposed procedure is simple and intuitive, with the crucial benefit of satisfying strong theoretical guarantees, there are also several unfortunate drawbacks associated with it. First, note that the update rule given in Equation equation 2 requires knowledge of the equilibrium policies of the benign agents in every iteration, which in turn requires knowledge of the Q-values of all agents, from every agent's point of view. This is a downside that all centralized, value-based, algorithms in MARL, such as Nash Q-Learning, share. Second, even if knowledge of the Q-values of all agents can be guaranteed, the problem of computing a Nash equilibrium from given utilities in a general-sum Markov game is known to be computationally hard (Daskalakis et al., 2009). Finally, the theoretical guarantees of the proposed method heavily rely on the stated assumptions. Such assumptions

may not always be satisfied in practice, where the irregularities in the individual utilities do not need to satisfy saddle point or global optima conditions. Motivated by the above, our next goal is thus to find a more practical and efficient approach to finding ARNEQ policies. In the next section, we introduce a model-free gradient-based algorithm that is able to empirically provide an efficient defense in various MARL environments.

## 4  Adversarially Robust Training for MARL

In this section, we propose an independent learning algorithm, Adversarially Robust Training (ART) for MARL, a gradient-based method that uses a two-timescale update rule, designed to provide a practically efficient defense against unknown adversarial policies.

Our robust notion of equilibrium imposes a specific training strategy. Note that the adversarial policies that complement an ARNEQ strategy profile are not necessarily identical across different players. Thus, each player, apart from computing its policy, also needs to compute its own set of adversarial agents and their associated adversarial policies. Thus, three different components need to be learned independently. Since each individual problem is a $\max\min$ problem, it is natural to consider the Gradient Descent Ascent (GDA) learning paradigm (Lin et al., 2020). This method has been widely used in the context of Markov games and has been shown to converge to equilibria in more structured regimes (Daskalakis et al., 2020).

Recall from the previous section that $\omega_i$, $\overline{\pi}_i$ and $\widehat{\pi}^i$ denote the adversarial subset selection policy, the benign policy, and adversarial joint policy with respect to agent $i$, respectively. We parametrize them as $\theta_i$, $\overline{\theta}_i$ and $\widehat{\theta^i}$, respectively. Based on these definitions, we define the loss with respect to a given adversarial subset selection policy $\omega_i$, dependent on joint policy $\overline{\pi}$ and adversarial policy $\widehat{\pi}^i$, as

$$\mathcal{L}_i(\theta_i, \overline{\theta}, \widehat{\theta^i}) = \mathbb{E}_{K \sim \omega_i(\theta_i)} \left[ V_i^\mu \left( \overline{\pi}_{-K}(\overline{\theta}_{-K}), \widehat{\pi}_K^i(\widehat{\theta_K^i}) \right) \right] \ .$$

Based on this definition, and the fact that we can write the above loss as an inner product, we have

$$\nabla_{\theta_i} \mathcal{L}_i(\theta_i, \overline{\theta}, \widehat{\theta^i}) = \left\langle \nabla_{\theta_i} \omega_i(\theta_i), V_i^\mu(\overline{\pi}(\overline{\theta}), \widehat{\pi}^i(\widehat{\theta^i})) \right\rangle \ ,$$

where $V_i^\mu(\overline{\pi}(\overline{\theta}), \widehat{\pi}^i(\widehat{\theta^i}))$ denotes the $\sum_{m=1}^k \binom{n-1}{m}$-dimensional vector with entries $V_i^\mu(\overline{\pi}_{-K}(\overline{\theta}_{-K}), \widehat{\pi}_K^i(\widehat{\theta_K^i}))$, for $K \in \mathcal{N}_i$.

The loss, with respect to player $i$, of the adversarial policy $\widehat{\pi}^i$, depending on $\omega_i$ and $\overline{\pi}$ is similarly defined. However, the gradient here is with respect to $\widehat{\theta^i}$. Note that, for each component $j$ of the adversarial policy $\widehat{\pi}^i$ the gradient $\nabla_{\widehat{\theta_j^i}} \mathcal{L}_i(\theta_i, \overline{\theta}, \widehat{\theta^i})$ is proportional to $\sum_{K : K \in \mathcal{N}_i \wedge j \in K} V_i^\mu(\overline{\pi}_{-K}(\overline{\theta}_{-K}), \widehat{\pi}_K^i(\widehat{\theta_K^i}))$, since for those subsets $K$ that do not contain $j$ as an adversarial component, the gradient becomes $0$.

Our learning protocol proceeds as follows. In every round $t \geq 1$, player $i \in \mathcal{N}$ maintains three gradient updates, one for the adversarial subset parameters $\theta_i^t$, one for the adversarial policy parameters $\widehat{\theta}^{i,t}$ and one for its own benign policy parameters $\overline{\theta}_i^t$. Furthermore, we set the same learning rate $\eta_A$ for both $\omega_i^{(t)}$ and $\widehat{\pi}_i^{(t)}$, since they both comprise the adversarial component of the problem.

At the beginning of the round, player $i$ collects samples of subsets from the adversarial subset selection policy of the previous round $\omega_i^{t-1}$. In order to perform the gradient updates of round $t$, player $i$ needs to collect several roll-outs. We denote by $\widetilde{\nabla} \mathcal{L}_i(\theta_i, \overline{\theta}, \widehat{\theta^i})$ the gradient estimate used for the updates based on the collected roll-outs.

The pseudocode of the described method is given in Algorithm 1.

## 5  Experimental Results

We evaluate policies trained using ART in two cooperative environments and one environment where the agents do not have aligned utilities. More specifically, we consider the Spread multi-agent-particle environment (Lowe

---

**Algorithm 1** ART for MARL (for player $i$)

---

**Input**: Number of potential adversaries $k$; accuracy parameter $\epsilon > 0$; learning rates $\eta_A$ and $\eta_B$; number of episodes $T$. **Initialize**: $\theta_i^0 = 0, \overline{\theta}^0 = 0, \widehat{\theta}_j^{i,0} = 0$, for all $i \in \mathcal{N}$, $j \neq i$.

1: **for** $t = 1, 2, \ldots, T$ **do**:
2:  Sample adversarial subsets from $\omega_i^{t-1}$.
3:  $\theta_i^t \leftarrow \theta_i^{t-1} - \eta_A \widetilde{\nabla}_{\theta_i} \mathcal{L}_i \left( \theta_i^{t-1}, \overline{\theta}^{t-1}, \widehat{\theta}^{i,t-1} \right)$
4:  $\overline{\theta}_i^t \leftarrow \overline{\theta}_i^{t-1} + \eta_B \widetilde{\nabla}_{\overline{\theta}_i} \mathcal{L}_i \left( \theta_i^t, \overline{\theta}^{t-1}, \widehat{\theta}^{i,t-1} \right)$
5:  **for** $j \in \mathcal{N}_i$ **do**:
6:    $\widehat{\theta}_j^{i,t} \leftarrow \widehat{\theta}_j^{i,t-1} - \eta_A \widetilde{\nabla}_{\widehat{\theta}_j^i} \mathcal{L}_i \left( \theta_i^t, \overline{\theta}^{t-1}, \widehat{\theta}^{i,t-1} \right)$
7:  **end for**
8: **end for**

---

et al., 2017), a cooperative multi-agent extension of the MuJoCo Ant environment (Todorov et al., 2012), and the cooperative Pursuit environment (Gupta et al., 2017). For detailed descriptions and parameters of the environments, see the Appendix (in the Supplementary Material).

| Environment | Algorithm | No Adversary | 1 Adversary | 2 Adversaries | 4 Adversaries |
|---|---|---|---|---|---|
| Independent Spread | Naive | $-\mathbf{34.0 \pm 1.9}$ | $-493.2 \pm 11.2$ | $-480.8 \pm 16.1$ | NA |
| | Fixed-K(k=1) | $-142.8 \pm 24.5$ | $-271.2 \pm 31.0$ | $-290.8 \pm 36.6$ | |
| | ART(k=1) | $-123.4 \pm 11.7$ | $-\mathbf{190.2 \pm 13.3}$ | $-242.0 \pm 33.4$ | |
| | ART(k=2) | $-148.0 \pm 15.1$ | $-201.4 \pm 11.7$ | $-\mathbf{213.6 \pm 9.9}$ | |
| Ant | Naive | $\mathbf{327.5 \pm 48.9}$ | $2.2 \pm 10.7$ | NA | NA |
| | Fixed-K(k=1) | $229.0 \pm 30.0$ | $29.5 \pm 14.8$ | | |
| | ART(k=1) | $181.9 \pm 25.4$ | $\mathbf{58.2 \pm 6.7}$ | | |
| Pursuit | Naive | $40.7 \pm 3.1$ | $34.4 \pm 1.2$ | $32.3 \pm 1.9$ | $-2.0 \pm 2.3$ |
| | Fixed-K(k=1) | $37.6 \pm 2.8$ | $35.7 \pm 4.1$ | $30.9 \pm 2.8$ | $8.5 \pm 2.7$ |
| | Fixed-K(k=2) | $29.2 \pm 10.5$ | $26.3 \pm 10.3$ | $25.7 \pm 9.8$ | $17.1 \pm 3.3$ |
| | Fixed-K(k=4) | $25.89 \pm 7.24$ | $23.82 \pm 7.50$ | $16.97 \pm 8.12$ | $6.21 \pm 6.27$ |
| | ART(k=1) | $\mathbf{47.3 \pm 1.3}$ | $\mathbf{50.6 \pm 1.8}$ | $\mathbf{48.1 \pm 1.9}$ | $\mathbf{26.0 \pm 4.3}$ |
| | ART(k=2) | $42.6 \pm 3.7$ | $37.4 \pm 1.7$ | $36.4 \pm 2.4$ | $21.0 \pm 4.0$ |
| | ART(k=4) | $35.6 \pm 4.3$ | $32.8 \pm 1.5$ | $34.0 \pm 1.6$ | $19.3 \pm 2.0$ |

Table 1: Comparison of ART with the proposed baselines. Fixed-K(k=2) for Independent Spread is omitted, as it is equivalent to ART(k=2). We report the mean and the standard error of the total reward achieved by an agent under attack over five runs with different seeds. In each run, we first train the benign agents and then train adversaries to minimize the total reward of one of the benign agents, considered to be a victim.

For Independent Spread, we report the worst-case total reward, where the minimum is taken over all possible victims and their adversaries. Since Ant and Pursuit are cooperative and symmetric environments, we consider an attacker that controls a specific subset of agents with cardinality $k$, while the remaining agents are victims who have a common reward function. The table reports the total reward of these agents. Note that, for Ant, we only consider $k = 1$, since an adversary that controls more agents can lift the ant up, denying any forward movement.

## 5.1 Baselines

We compare the effectiveness of ART with Naive Training, where no adversarial agents are present during the training phase. This provides a baseline for evaluating the resilience of robustly trained agents compared to conventional techniques. Additionally, we introduce the Fixed-K training baseline where the subset of agents controlled by the adversary is fixed during the training process. We aim to signify the importance of training with the worst-case subset of adversaries. Both baselines are trained with the same hyperparameters as ART, ensuring a fair comparison across the board.

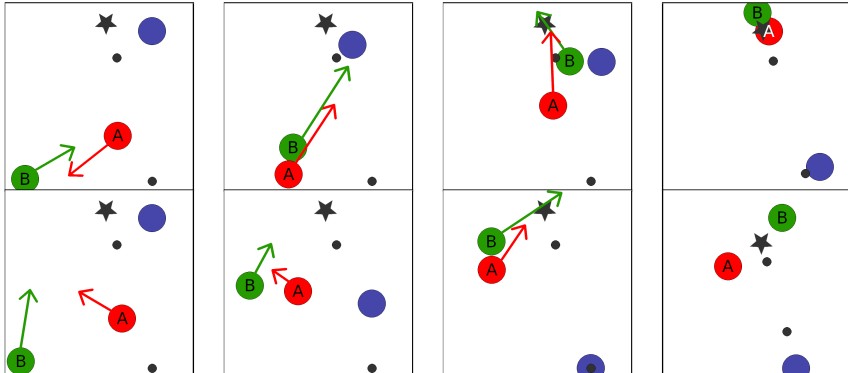

Figure 2: Consecutive snapshots of agents under attack by one adversary in the Independent Spread environment: first row represents Naive Training and second row represents ART(k=1). The green circle corresponds to the agent under attack, red to the adversary, blue to the benign agent not under attack, the black star is the target landmark of the attacked agent, and the black circles are the target landmarks of other agents. Naively trained agents do not avoid other agents and allow the adversary to infer the position of their landmark, resulting in the adversary blocking it. Agents trained with ART choose a longer route to their destination, which avoids adversarial agents, and maintain distance around their landmark, which does not allow the adversary to infer its exact location.

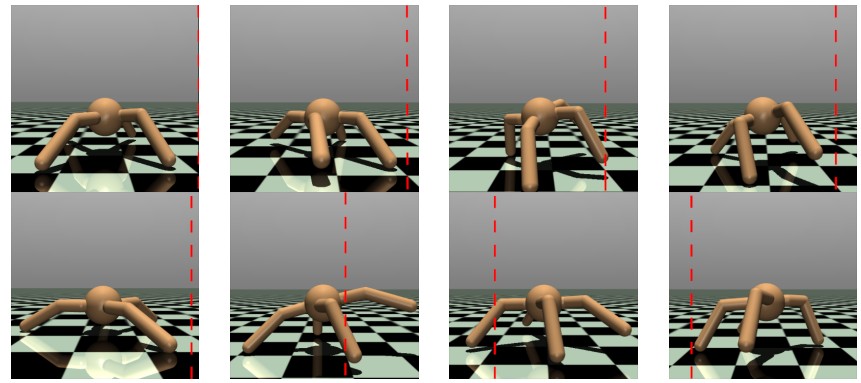

Figure 3: Consecutive snapshots of agents under attack in the Ant environment: first row represents naive training and second row represents ART(k=1). The dotted red line in Ant serves as a stationary reference point. Naively trained agents struggle to move forward when under attack. Robustly trained ones are able to move forward when one leg is corrupted.

## 5.2 Implementation Details

To train robust neural policies, we implemented ART as described in Algorithm 1. Both the adversarial and benign policies are updated using the PPO (Schulman et al., 2017) implementation of the *ray* library (Moritz et al., 2018). The adversarial subset selection model $\omega$, as described in Section 4 is implemented using *pytorch* (Paszke et al., 2019) and utilizes REINFORCE (Sutton et al., 1999) update rules. To allow training $\omega$ with more data, we used additionally generated trajectories. Hyperparameters were selected according to the fine-tuned examples provided in the *ray* library, except for the learning rate of benign agents, which is set to be half of the learning rate of adversarial agents. This leads to aggressively updated adversaries, while still ensuring convergence in a reasonable time. All used hyperparameters can be found in the Appendix (see the Supplementary Material).

## 5.3 Empirical Analysis

**Quantitative analysis.** To evaluate the robustness of the obtained agents, we fixed the benign policies and trained new adversaries from scratch that use the same hyperparameters. We used the latter to test if the robust policies generalize to new attackers. Table 1 reports the test-time results of ART and both baselines. We observe that in all environments, ART consistently outperforms Naive Training when under

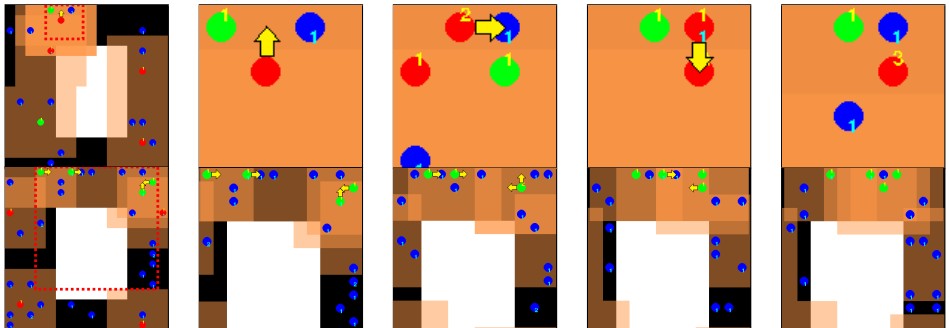

Figure 4: Consecutive snapshots of agents under attack by 4 adversaries in the Pursuit environment: the first row is a result of Naive Training and the second row is a result of ART(k=1). For better visibility, we zoomed in on the area of interest (indicated by the red dotted area). A full version can be found in the Appendix (see the Supplementary Material). Red circles correspond to adversarial agents, green to benign agents, and blue to evaders, the orange boxes to the field of view of agents. Yellow arrows show the movement of the adversary in the upper row and benign agents in the lower row. Naively trained agents expect cooperation, leading to agents being easily fooled. The adversary moves upward instead of left, resulting in the prey not being captured. Agents trained with ART learned to effectively avoid contact with other agents, until they arrive at a specific location. This technique ensures that in every group of agents, there are always enough benign agents present to fully surround prey.

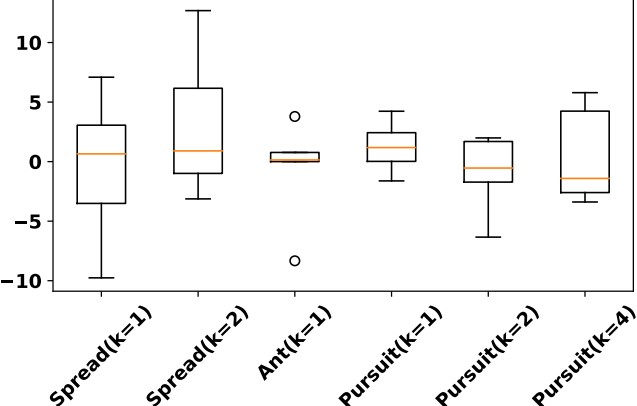

Figure 5: Achievable increase in the performance of a single agent, i.e., victim, after we train the benign agents and adversaries as described in Table 1. We consider one fixed victim agent and 5 different runs with different seeds. In each run, we train the benign policies until convergence using ART. Afterward, we select one agent and train an adversary attacking the selected agent. Then, training of the selected agent is continued for 5% of the original number of training steps, while keeping all other policies fixed. Depicted is the difference between the performance of the improved policy and the original performance. If the players have converged to an ARNEQ, then further training a given agent when the rest of the players keep playing fixed strategies should not imply increased performance for this player. Otherwise, this would mean there is room for improvement which would violate the very condition of an equilibrium. Note that, on average, performance tends to stay the same, except for some small oscillations which are due to the stochasticity of deep RL methods. This suggests that players are close to an ARNEQ.

attack. Fixed-K training results in a more robust policy than naive training, but the benign agents tend to overfit to the adversaries present during training, resulting in worse performance when under attack by different adversarial agents. The importance of anticipating the correct number of adversaries during the training procedure is environment-specific. In Independent Spread, optimal performance of ART is achieved if the number of adversaries during training matches their number at deployment time. In pursuit, training with a smaller number of adversaries yields better performance in all scenarios. We conjecture that this may be due to the hardness of training with a larger number of adversarial agents. In the Independent Spread and Ant environment the resilience against adversarial attacks of ART and Fixed-K comes at the cost of decreased performance in a clean environment. In Pursuit, adversarial training is beneficial even when there

is no attack. This can, in part, be explained by examining the behavior of the trained agents, which we discuss in the qualitative analysis.

**Qualitative analysis.** Next, we analyze how the behavior of agents trained with ART differs from naive training in the examined environments. For Independent Spread, Figure 2 shows that agents trained using ART opt for longer routes, strategically avoiding potential adversaries. This behavior explains the observed difference in performance when comparing ART and naive training in clean and adversarial environments. For Ant, Figure 3 depicts the ability of ART to train agents that are able to move the ant forward even if one leg is acting maliciously. As can be seen in the figure, we do not learn such a robust policy with Naive Training. For Pursuit, Figure 4 depicts that agents trained using ART meet up at a specific point at the beginning of the episode. This strategic rendezvous not only allows them to navigate the environment in a trusted group, enabling them to bypass interactions with adversaries but seems to contribute to improved performance even when not under attack.

**Convergence.** Finally, we aim to evaluate the convergence of ART by analyzing the achievable increase in performance of a single agent. A small increase signifies the approximate convergence of ART to an ARNEQ. Figure 5 depicts this increase in performance for all evaluated environments and differing numbers of adversaries. As the achieved improvement is small, this suggests that we approach an ARNEQ.

## 6 Concluding Discussion

We considered the problem of defending against unknown adversarial policies in general-sum Markov games. We proposed a novel solution concept, namely, the adversarially robust Nash equilibrium (ARNEQ). Further, we provided a theoretical characterization of our solution concept and analyzed a centralized method for computing it. We also proposed a practical decentralized algorithm to compute an ARNEQ and empirically demonstrated its efficiency across three different environments.

On the experimental front, our paper provides comprehensive results in three different multi-agent reinforcement learning environments, all of which showcase the benefit of using our defense algorithm at training. Conducting further experiments on more challenging environments with more agents would provide further insight into the efficiency of our method. The nature of our algorithm necessitates the simultaneous computation of as many estimates as there are players, for each individual player. Thus, it would be interesting to see how scalable our method is for large games. Here, approaches such as mean-field multi-agent reinforcement learning might provide essential tools.

On the theoretical front, we provided characterization results for convergence in a centralized setting of our model. A key open question that is left in the paper is whether we can restrict the attack model so that we are able to provide theoretical convergence guarantees on independent gradient-based learning methods in more structured regimes. For instance, in Markov potential games, it is known that independent learning converges to an approximate Nash equilibrium due to a common potential function for all players. Under our attack model, this game structure is damaged, since the players constantly update their estimates of utilities for the other players at training time – which are different from the original utilities of the underlying game. Thus, a natural research question would be: can we utilize the underlying potential function, and characterize the quantity of deviation of the modified value functions from their original values using this potential function? Addressing such a question is an interesting future research direction.

### Acknowledgments

This research was, in part, funded by the Deutsche Forschungsgemeinschaft (DFG, German Research Foundation) – project number 467367360.

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

# Appendix

## Table of Contents

## A  Proof of Existence (Theorem 1)

For the existence proof, we will use the Kakutani's fixed point Theorem. This result has been classically used in proving the existence of Nash equilibria in various types of Markov games. It requires the careful construction of a set-valued function whose fixed point would represent the equilibrium of the game of interest. Once the construction is made, the theorem states that, if such a function satisfies some technical conditions, then the existence of its fixed point is guaranteed, thus effectively proving the existence of an equilibrium of the game. This will be our approach in the following. First, we will prove some auxiliary results related to properties such as contraction, continuity, and convexity. Then, we will construct a set-valued function whose fixed point would represent an ARNEQ and further show that it satisfies the technical conditions of Kakutani's fixed point theorem.

### A.1  Auxiliary Results

In this section we prove some auxiliary results which will be needed in the proof of Theorem 1.

We will use the following definitions only for this section. Fix an arbitrary state $s$ and player $i$. Let $x_s^i$ denote a benign policy $\overline{\pi}_i(\cdot|s)$ of player $i$ at state $s$, and let $y_s^{-i}$ denote the joint policy $\overline{\pi}_{-i}(\cdot|s)$ of the rest of the benign players. Moreover, let $z_s^i$ denote an adversarial joint policy $\widehat{\pi}_i(\cdot|s)$ and $\omega_s^i$ denote the randomized strategy $\omega_i$, but only applied on state $s$. We also denote by $\mathcal{X}$ the policy space $\Delta(\mathcal{A}_i)$ and by $\mathcal{Y}$ the joint policy space $\times_{j \neq i}\Delta(\mathcal{A}_j)$. Note that we do not assume action spaces that depend on the state. Let $\mathcal{Z} = \mathcal{X} \times \mathcal{Y}$ denote the whole policy space $\Pi$.

Now, given a vector of value functions $(V^i)_{i \in \mathcal{N}}$ in some compact value function space $\mathcal{V} = \mathcal{V}_1 \times \ldots \times \mathcal{V}_n$, let us define, for each player $i \in \mathcal{N}$ and state $s \in \mathcal{S}$, the function

$$f_s^i\left(x_s^i, y_s^{-i}, z_s^i, \omega_s^i, V^i\right) := \mathbb{E}_{a \sim (x_s^i, y_s^{-i}, z_s^i), \omega_s^i}\left[\mathcal{R}(s, a) + \gamma\left\langle\mathcal{P}(s, a), V^i\right\rangle\right] \ .$$

Next, let us define

$$\phi_s^i(x_s^i, y_s^{-i}, V^i) := \min_{z_s^i \in \mathcal{Y}} \min_{\omega_s^i \in \Omega_i} f_s^i\left(x_s^i, y_s^{-i}, z_s^i, \omega_s^i, V^i\right) \ ,$$

and

$$\psi_s^i(y_s^{-i}, V^i) := \max_{x_s^i \in \mathcal{X}} \phi_s^i(x_s^i, y_s^{-i}, V^i) \ .$$

Note that, the spaces $\mathcal{X}$, $\mathcal{Y}$, and $\Omega_i$ are compact spaces (closed and bounded) as policy spaces. Our first result shows that the mapping $\psi$ is a contraction mapping on the value function spaces.

**Lemma 1** *Given joint policy $x$, let $\psi_x : \mathcal{V} \to \mathcal{V}$, be such that, for each $i, s$ we have $\psi_x(V) = (\psi_s^i(y_s^{-i}, V^i))_{i,s}$. Then, the mapping $\psi$ is a contraction mapping.*

**Proof:** Let $V, U \in \mathcal{V}$. Fix a player $i$ and state $s$. Note that we have

$$
\begin{aligned}
\left| \psi_s^i(y_s^{-i}, V^i) - \psi_s^i(y_s^{-i}, U^i) \right| &= \left| \max_{x_s^i \in \mathcal{X}} \phi_s^i(x_s^i, y_s^{-i}, V^i) - \max_{x_s^i \in \mathcal{X}} \phi_s^i(x_s^i, y_s^{-i}, U^i) \right| \\
&\leq \left| \phi_s^i(\widetilde{x}_s^i, y_s^{-i}, V^i) - \phi_s^i(\widetilde{x}_s^i, y_s^{-i}, U^i) \right| \\
&= \left| \min_{z_s^i \in \mathcal{Y}} \min_{\omega_s^i \in \Omega_i} f_s^i\left(\widetilde{x}_s^i, y_s^{-i}, z_s^i, \omega_s^i, V^i\right) - \min_{z_s^i \in \mathcal{Y}} \min_{\omega_s^i \in \Omega_i} f_s^i\left(\widetilde{x}_s^i, y_s^{-i}, z_s^i, \omega_s^i, U^i\right) \right| \\
&\leq \left| f_s^i\left(\widetilde{x}_s^i, y_s^{-i}, \widetilde{z}_s^i, \widetilde{\omega}_s^i, V^i\right) - f_s^i\left(\widetilde{x}_s^i, y_s^{-i}, \widetilde{z}_s^i, \widetilde{\omega}_s^i, U^i\right) \right| \\
&= \left| \mathbb{E}_{a \sim (\widetilde{x}_s^i, y_s^{-i}, \widetilde{z}_s^i, \widetilde{\omega}_s^i)} \left[ \left( \mathcal{R}(s,a) + \gamma \left\langle \mathcal{P}(s,a), V^i \right\rangle \right) - \left( \mathcal{R}(s,a) + \gamma \left\langle \mathcal{P}(s,a), U^i \right\rangle \right) \right] \right| \\
&= \left| \mathbb{E}_{a \sim (\widetilde{x}_s^i, y_s^{-i}, \widetilde{z}_s^i, \widetilde{\omega}_s^i)} \left[ \gamma \left\langle \mathcal{P}(s,a), V^i - U^i \right\rangle \right] \right| \\
&\leq \gamma \left\| V^i - U^i \right\|_\infty \ ,
\end{aligned}
$$

where for the first inequality, we let $\widetilde{x}_s^i$ be the maximizer of $\phi_s^i(x_s^i, y_s^{-i}, V^i)$, for the second inequality we let $(\widetilde{z}_s^i, \widetilde{\omega}_s^i)$ be the minimizer of $f_s^i\left(\widetilde{x}_s^i, y_s^{-i}, z_s^i, \omega_s^i, U^i\right)$, and for the last inequality we use Hölder. $\qquad\square$

Next, we will show that there exists a unique robust value vector of the above mapping, a result that follows from Banach's contraction theorem which we state below.

**Theorem 3 (Banach's Contraction Mapping Theorem)** *Let $(\mathcal{V}, \|\cdot\|)$ be a complete metric space and let $\psi : \mathcal{V} \to \mathcal{V}$ be a contraction mapping. Then there exists a unique fixed point of the function $\psi$.*

**Lemma 2** *For any given joint policy $x$, player $i \in \mathcal{N}$ and state $s \in \mathcal{S}$, there exists a unique value vector $V^i$ such that*

$$V_s^i = \max_{x_s^i \in \mathcal{X}} \min_{z_s^i \in \mathcal{Y}} \min_{\omega_s^i \in \Omega_i} f_s^i\left(x_s^i, y_s^{-i}, z_s^i, \omega_s^i, V^i\right) = \psi_s^i(y_s^{-i}, V^i) \ .$$

**Proof:** Note that, the fact that the metric space $(\mathcal{V}, \|\cdot\|_\infty)$ is complete, since $\mathcal{V}$ is compact, together with Lemma 1 above, implies the desired result as a consequence of Banach's contraction mapping point theorem (Theorem 3). $\qquad\square$

Now we need to show that our functions $f_s^i$ are equicontinuous. First, let us define the following norm jointly on policies and value functions. Given joint policies $x$ and $y$, and value functions $V$ and $U$, for every state $s$ and player $i$, let

$$d\left((x_s, V^i), (y_s, U^i)\right) = \max_{i \in \mathcal{N}, a \in \mathcal{A}_i} \left| x_{sa}^i - y_{sa}^i \right| + \max_{s \in \mathcal{S}} \left| V_s^i - U_s^i \right| \ ,$$

where $x_s$ denotes the joint policy vector for state $s$, $x_{sa}^i$ denotes the probability of player $i$ taking action $a \in \mathcal{A}_i$ in state $s$, while $V_s^i$ denotes the value of state $s$ for player $i$ with respect to value vector $V$.

**Lemma 3** *Fix $s \in \mathcal{S}$ and player $i$. Given $\epsilon > 0$, there exists $\delta(\epsilon) > 0$, such that, if for any $\boldsymbol{p} = (x_s, V^i)$ and $\boldsymbol{q} = (y_s, U^i)$, we have $d(\boldsymbol{p}, \boldsymbol{q}) < \delta(\epsilon)$, then, for all $z_s^i \in \mathcal{Y}$ and $\omega_s^i \in \Omega_i$, we have*

$$\left| f_s^i(x_s, z_s^i, \omega_s^i, V^i) - f_s^i(y_s, z_s^i, \omega_s^i, U^i) \right| < \epsilon \ .$$

**Proof:** First, note that, since rewards are in the unit interval, value functions are bounded by $1/(1-\gamma)$. Letting $\boldsymbol{p} = (x_s, V)$ and $\boldsymbol{q} = (y_s, U)$, we have

$$
\left| f_s^i(x_s, z_s^i, \omega_s^i, V^i) - f_s^i(y_s, z_s^i, \omega_s^i, U^i) \right|
$$

$$
= \left| \mathbb{E}_{a\sim(x_s, z_s^i, \omega_s^i)} \left[ \mathcal{R}(s,a) + \gamma \langle \mathcal{P}(s,a), V^i \rangle \right] - \mathbb{E}_{a\sim(y_s, z_s^i, \omega_s^i)} \left[ \mathcal{R}(s,a) + \gamma \langle \mathcal{P}(s,a), U^i \rangle \right] \right|
$$

$$
\leq \mathbb{E}_{K\sim\omega_i} \left[ \left| \sum_{a\in\mathcal{A}} \prod_{j\in K} z_{sa_j}^{i,j} \prod_{l\notin K} x_{sa_l}^i \left( \mathcal{R}(s,a) + \gamma \langle \mathcal{P}(s,a), V^i \rangle \right) \right. \right.
$$

$$
\left. \left. - \sum_{a\in\mathcal{A}} \prod_{j\in K} z_{sa_j}^{i,j} \prod_{l\notin K} y_{sa_l}^i \left( \mathcal{R}(s,a) + \gamma \langle \mathcal{P}(s,a), U^i \rangle \right) \right| \right]
$$

$$
\leq \left| \sum_{a\in\mathcal{A}} \prod_{j\in K^*} z_{sa_j}^{i,j} \prod_{l\notin K^*} x_{sa_l}^i \left( \mathcal{R}(s,a) + \gamma \langle \mathcal{P}(s,a), V^i \rangle \right) \right.
$$

$$
\left. - \sum_{a\in\mathcal{A}} \prod_{j\in K^*} z_{sa_j}^{i,j} \prod_{l\notin K^*} y_{sa_l}^i \left( \mathcal{R}(s,a) + \gamma \langle \mathcal{P}(s,a), U^i \rangle \right) \right|
$$

$$
\leq \sum_{a\in\mathcal{A}} \left| \prod_{j\in K^*} z_{sa_j}^{i,j} \left( \prod_{l\notin K^*} x_{sa_l}^i - \prod_{l\notin K^*} y_{sa_l}^i \right) \mathcal{R}(s,a) \right|
$$

$$
+ \gamma \sum_{a\in\mathcal{A}} \left| \prod_{j\in K^*} z_{sa_j}^{i,j} \left( \prod_{l\notin K^*} x_{sa_l}^i - \prod_{l\notin K^*} y_{sa_l}^i \right) \langle \mathcal{P}(s,a), V^i \rangle \right|
$$

$$
+ \gamma \sum_{a\in\mathcal{A}} \left| \prod_{j\in K^*} z_{sa_j}^{i,j} \prod_{l\notin K^*} y_{sa_l}^i \langle \mathcal{P}(s,a), V^i - U^i \rangle \right|
$$

$$
\leq \sum_{a\in\mathcal{A}} \left| \prod_{j\in K^*} z_{sa_j}^{i,j} \left( \prod_{l\notin K^*} x_{sa_l}^i - \prod_{l\notin K^*} y_{sa_l}^i \right) \right| + \frac{\gamma}{1-\gamma} \sum_{a\in\mathcal{A}} \left| \prod_{j\in K^*} z_{sa_j}^{i,j} \left( \prod_{l\notin K^*} x_{sa_l}^i - \prod_{l\notin K^*} y_{sa_l}^i \right) \right|
$$

$$
+ \gamma \sum_{a\in\mathcal{A}} \left| \prod_{j\in K^*} z_{sa_j}^{i,j} \prod_{l\notin K^*} y_{sa_l}^i \left\| V^i - U^i \right\|_\infty \right|
$$

$$
= \frac{1}{1-\gamma} \sum_{a\in\mathcal{A}} \left| \prod_{j\in K^*} z_{sa_j}^{i,j} \left( \prod_{l\notin K^*} x_{sa_l}^i - \prod_{l\notin K^*} y_{sa_l}^i \right) \right| + \gamma A^n \left\| V^i - U^i \right\|_\infty \;,
$$

where the second inequality follows from Hölder's inequality, where $K^*$ denotes the set that maximizes the difference inside the absolute value; for the third inequality we have just used an algebraic artifice, adding and subtracting a term related to $y^i$ and $V^i$, and then rearranging; the third inequality uses the fact that the rewards are in the unit interval, and Hölder's inequality.

Now, let us define

$$
\delta_1(\epsilon) = \frac{(1-\gamma)\min\{\epsilon,1\}}{2(2^{n-k}-1)A^n}, \quad \text{and} \quad \delta_2(\epsilon) = \frac{\min\{\epsilon,1\}}{2\gamma A^n} \;.
$$

Moreover, let $\alpha_{s,a_i}^i = y_{sa_i}^i - x_{sa_i}^i$. Assume that $|\alpha_{sa_i}^i| < \min\{\delta_1(\epsilon), \delta_2(\epsilon)\}$, and $|V_s^i - U_s^i| < \min\{\delta_1(\epsilon), \delta_2(\epsilon)\}$ for any player $i \in \mathcal{N}$, state $s \in \mathcal{S}$ and action $a \in \mathcal{A}$. For the first term on the right-hand-side of the last equality above, we have

$$
\frac{1}{1-\gamma} \sum_{a\in\mathcal{A}} \left| \prod_{j\in K^*} z_{sa_j}^{i,j} \left( \prod_{l\notin K^*} x_{sa_l}^i - \prod_{l\notin K^*} y_{sa_l}^i \right) \right|
$$

$$\leq \frac{1}{1-\gamma} \sum_{a \in \mathcal{A}} \prod_{j \in K^*} z_{sa_j}^{i,j} \sum_{\substack{M \subset \mathcal{N} \setminus K^* \\ |M| \geq 1}} \left| \prod_{m \in M} \alpha_{s,a_m}^m \right| \left| \prod_{m \notin M^C} y_{sa_m}^m \right|$$

$$\leq \frac{1}{1-\gamma} \sum_{a \in \mathcal{A}} \prod_{j \in K^*} z_{sa_j}^{i,j} \sum_{\substack{M \subset \mathcal{N} \setminus K^* \\ |M| \geq 1}} \left| \prod_{m \in M} \alpha_{s,a_m}^m \right|$$

$$\leq \frac{1}{1-\gamma} \sum_{a \in \mathcal{A}} \sum_{\substack{M \subset \mathcal{N} \setminus K^* \\ |M| \geq 1}} \left| \alpha_{s,a_{m'}}^{m'} \right|$$

$$\leq \frac{\epsilon}{2} \,,$$

where the first inequality follows from the algebraic identity

$$\left| \prod_{m=1}^{n-k} (y_{sa_m}^m + \alpha_{sa_m}^m) - \prod_{m=1}^{n-k} y_{sa_m}^m \right| = \left| \sum_{\substack{M \subseteq \mathcal{N} \setminus K^* \\ |M| \geq 1}} \left( \prod_{m \in M} \alpha_{sa_m}^m \right) \left( \prod_{m \in M^C} y_{sa_m}^m \right) \right| \,,$$

the second inequality follows from the fact that the product of numbers in the unit interval is no greater than 1; the third inequality uses a similar argument on the policies $z_{sa_j}^j$ and the rest of the differences $\alpha_{sa_m}^m$, except $\alpha_{sa_{m'}}^{m'}$, where we note that $\alpha_{sa_m}^m$ are also in the unit interval; the last inequality follows by definition of $\delta_1(\epsilon)$. On the other hand, note that we also have

$$\gamma A^n \left\| V^i - U^i \right\|_\infty < \frac{\epsilon}{2} \,,$$

by definition of $\delta_2(\epsilon)$. Thus, we obtain that

$$\left| f_s^i(x_s, z_s^i, \omega_s^i, V^i) - f_s^i(y_s, z_s^i, \omega_s^i, U^i) \right| < \frac{\epsilon}{2} + \frac{\epsilon}{2} = \epsilon \,.$$

$\square$

Next, we will state the following lemmas, which will be the rest of the necessary ingredients for the existence proof. We omit their proofs, since they follow the same lines as in (Kardeş et al., 2011; Fink, 1964). First, let us define the unique best response for player $i$ as

$$\tau^i(y^{-i}) = \left\{ V^i : V_s^i = \max_{x_s^i} \min_{z_s^i} \min_{\omega_s^i} f_s^i(x_s^i, y_s^{-i}, z_s^i, \omega_s^i, V^i), \forall s \in \mathcal{S} \right\} \,.$$

**Lemma 4** *The function $\phi_s^i(x_s^i, y_s^{-i}, V^i)$ is continuous in all its variables, for all $i \in \mathcal{N}$ and $s \in \mathcal{S}$.*

**Proof:** The result immediately follows from the fact that the pointwise minimum of a family of equicontinuous functions is continuous. $\square$

**Lemma 5** *The function $\phi_s^i(x_s^i, y_s^{-i}, V^i)$ is concave in $x_s^i$, for a fixed $y_s^{-i}$ and $V^i$.*

**Proof:** The result immediately follows by definition of $\phi_s^i$. $\square$

**Lemma 6** *The function $\psi_s^i(y_s^{-i}, V^i)$ is continuous in $y_s^{-i}$. Furthermore, the set $\{\psi_s^i(y_s^{-i}, V^i) | V^i \text{ is bounded}\}$ is equicontinuous.*

**Proof:** The result follows from Lemma 3 above and Lemma 3 in (Fink, 1964). $\square$

**Lemma 7** *If the sequence $y^{-i,n}$ goes to $y^{-i}$ and $\tau_s^i(y^{-i,n})$ goes to $V_s^i$, as $n$ goes to $\infty$, then*

$$\tau_s^i(y_s^{-i}) = V_s^i \,.$$

**Proof:** Observe that

$$\left| V_s^i - \psi_s^i(y_s^{-i,n}, V^i) \right| \leq \left| V_s^i - \tau_s^i(y^{-i,n}) \right| + \left| \tau_s^i(y^{-i,n}) - \psi_s^i(y_s^{-i}, \tau_s^i(y^{-i,n})) \right| + \left| \psi_s^i(y_s^{-i}, \tau_s^i(y^{-i,n})) - \psi_s^i(y_s^{-i,n}, V^i) \right| ,$$

for every $n \geq 1$, $i \in \mathcal{N}$ and $s \in \mathcal{S}$. Now note that $\left| V_s^i - \tau_s^i(y^{-i,n}) \right| \to 0$ and $\left| \psi_s^i(y_s^{-i}, \tau_s^i(y^{-i,n})) - \psi_s^i(y_s^{-i,n}, V^i) \right| \to 0$, as $n \to \infty$, by assumption. Moreover, Lemma 6 implies that $\left| \tau_s^i(y^{-i,n}) - \psi_s^i(y_s^{-i}, \tau_s^i(y^{-i,n})) \right| = \left| \psi_s^i(y_s^{-i}, \tau_s^i(y^{-i,n})) - \psi_s^i(y_s^{-i}, \tau_s^i(y^{-i,n})) \right| \to 0$, as $n \to \infty$. Hence, $\left| V_s^i - \tau_s^i(y^{-i,n}) \right| \to 0$, as $n \to \infty$. $\qquad\square$

## A.2 The Proof of Theorem 1

In this section, we will conclude the proof of Theorem 1. In order to do that, we will make use of the famous Kakutani's fixed point theorem which we state below. First, let us define the notion of upper semi-continuous functions, which is a precondition of this result.

**Definition 4** *A correspondence $\kappa : \mathcal{Z} \to 2^{\mathcal{Z}}$ is said to be upper semicontinuous if $y_n \in \kappa(x_n)$, for all $n \geq 1$, $\lim_{n \to \infty} x_n = x$ and $\lim_{n \to \infty} y_n = y$ imply that $y \in \kappa(x)$.*

With this, we can now state Kakutani's fixed point theorem.

**Theorem 4** *(Kakutani's fixed point theorem) If $\mathcal{Z}$ is a closed, bounded and convex set in a Euclidean space, and $\kappa$ is an upper semicontinuous correspondence mapping $\mathcal{Z}$ into the family of closed convex subsets of $\mathcal{Z}$, then there exists $x \in \mathcal{Z}$ such that $x = \kappa(x)$.*

We will construct a correspondence that satisfies the conditions of the theorem, and show that its fixed point is an equilibrium point. The correspondence we need is the following:

$$\kappa(x) = \left\{ y \in \mathcal{Z} : y_s^i \in \arg\max_{u_s^i \in \mathcal{X}} \phi_s^i(u_s^i, x_s^{-i}, V^i), \; V_s^i = \max_{u_s^i \in \mathcal{X}} \phi_s^i(u_s^i, x_s^{-i}, V^i), \forall s \in \mathcal{S}, i \in \mathcal{N} \right\}$$

In the previous section, we have shown that the functions $\phi_s^i$ satisfy continuity. We will further show that the defined set function $\kappa$ satisfies the conditions of Kakutani's theorem. Let us first restate Theorem 1 for convenience, and then proceed to its proof.

**Statement 1** *Let $\mathcal{G}$ be a finite game with $n$ players and let $k < n$. Assume that an arbitrary subset of at most $k$ players use arbitrarily modified utilities, and assume that the mixed strategies of all players lie in compact sets. Also, assume that, for each player $i \in \mathcal{N}$, the sets $\Omega_i$ are compact. Then, a $k$-ARNEQ, as given in Definition 2 exists.*

**Proof:** We will use the same argument as in the proof of Theorem 4 of (Kardeş et al., 2011). We repeat the argument here with our notation for completion.

Given $i \in \mathcal{N}$, $s \in \mathcal{S}$, Lemma 4 shows that the function $\phi_s^i(x_s^i, y_s^{-i}, V^i)$ is continuous in all of its variables. This, together with the fact that its domain is compact, implies that $\phi_s^i$ achieves its maximum, that is $\arg\max_{u_s^i \in \mathcal{X}} \phi_s^i(u_s^i, x_s^{-i}, V^i) \neq \emptyset$. On the other hand, Theorem 2 implies that $V_s^i = \max_{x_s^i \in \mathcal{X}} \phi_s^i(x_s^i, y_s^{-i}, V^i)$. Thus, we have that $\kappa(x) \neq \emptyset$, by definition of $\kappa$ above.

Next, we show that $\kappa(x)$ is a convex set. Suppose that $u, v \in \kappa(x)$, for some $u = (u^1, \ldots, u^n)$ and $v = (v^1, \ldots, v^n)$. Then, for any $y \in \mathcal{Z}$, $i \in \mathcal{N}$ and $s \in \mathcal{S}$, we have, by definition of $\kappa(x)$, that

$$V_s^i = \phi_s^i(u_s^i, x_s^{-i}, V^i) = \phi_s^i(v_s^i, x_s^{-i}, V^i) \geq \phi_s^i(y_s^i, x_s^{-i}, V^i) ,$$

for any $y \in \mathcal{Z}$. Thus, for any $\lambda \in [0,1]$, $i \in \mathcal{N}$, $s \in \mathcal{S}$, and by concavity of $\phi_s^i(y_s^i, x_s^{-i}, V^i)$, we have

$$\phi_s^i(y_s^i, x_s^{-i}, V^i) \leq V_s^i = \lambda \phi_s^i(u_s^i, x_s^{-i}, V^i) + (1-\lambda)\phi_s^i(v_s^i, x_s^{-i}, V^i)$$
$$\leq \phi_s^i\left((\lambda u_s^i + (1-\lambda)v_s^i), x_s^{-i}, V^i\right)$$

$$\leq V_s^i \ ,$$

which implies that $\lambda u_s^i + (1 - \lambda)v_s^i \in \kappa(x)$. Thus, the image of $\kappa$ is convex.

Next, we will show that $\kappa$ is upper semicontinuous. To that end, suppose that $x^n \to x$ and $y^n \to y$, and that $y^n \in \kappa(x^n)$, for $n \geq 1$. Note that $\tau_s^i(x_s^{-i,n}) \to V_s^i$, since $x^{-i,n} \to x^{-i}$, as a subsequence of $(x^n)_{n \geq 1}$. Using the triangle inequality, we obtain

$$\left| \phi_s^i \left( y_s^i, x_s^{-i}, V^i \right) - V^i \right|$$
$$\leq \left| \phi_s^i \left( y_s^i, x_s^{-i}, V^i \right) - \phi_s^i \left( y_s^{i,n}, x_s^{-i,n}, \tau_s^i(x^{-i,n}) \right) \right| + \left| \phi_s^i \left( y_s^{i,n}, x_s^{-i,n}, \tau_s^i(x^{-i,n}) \right) - V_s^i \right|$$
$$= \left| \phi_s^i \left( y_s^i, x_s^{-i}, V^i \right) - \phi_s^i \left( y_s^{i,n}, x_s^{-i,n}, \tau_s^i(x^{-i,n}) \right) \right| + \left| \tau_s^i(x^{-i,n}) - V_s^i \right| \to 0, \text{ as } n \to \infty$$

Thus, $V_s^i = \phi_s^i(y_s^i, x_s^{-i}, V^i)$. Lemma 7 also implies that $\tau_s^i(x^{-i}) = V_s^i$. Thus, we obtain

$$V_s^i = \phi_s^i(y_s^i, x_s^{-i}, V^i) = \tau_s^i(x^{-i}) = \max_{u_s^i \in \mathcal{X}} \phi_s^i(u_s^i, x_s^{-i}, V^i) \ .$$

Therefore, $y \in \kappa(x)$, which is what we needed, in order to prove that $\kappa$ is upper semicontinuous. The fact that $\kappa(x)$ is a closed set for any $x \in \mathcal{X}$ follows by definition of upper semicontinuity. Thus, $\kappa$ satisfies the conditions of Theorem 4, which implies that its fixed point exists, that is, there exists joint policies $x = (x^1, \dots, x^n)$ and value functions $V^1, \dots, V^n)$, such that, for any $i \in \mathcal{N}$ and $s \in \mathcal{S}$, we have

$$x_s^i \in \arg \max_{u_s^i \in \mathcal{X}} \min_{\omega_s^i \in \Omega_i} \min_{z_s^i \in \mathcal{Y}} \mathbb{E}_{a \sim x_s^i, x_s^{-i}, z_s^i, \omega_s^i} \left[ \mathcal{R}(s, a) + \gamma \left\langle \mathcal{P}(s, a), V^i \right\rangle \right] \ ,$$

and

$$V_s^i = \max_{u_s^i \in \mathcal{X}} \min_{\omega_s^i \in \Omega_i} \min_{z_s^i \in \mathcal{Y}} \mathbb{E}_{a \sim x_s^i, x_s^{-i}, z_s^i, \omega_s^i} \left[ \mathcal{R}(s, a) + \gamma \left\langle \mathcal{P}(s, a), V^i \right\rangle \right] \ .$$

Thus, an ARNEQ of $\mathcal{G}$ exists. $\qquad\square$

## B Proof of Convergence of Adversarially Robust Nash Q-Learning (Theorem 2)

In this section, we will formally prove that the Nash Q-Learning approach to Adversarially Robust Training convergence to an ARNEQ under certain technical assumptions. We first define some necessary additional notions and then state the assumptions.

Given $t \geq 1$, let $(\overline{Q}_i^t(s))_{i \in \mathcal{N}}$ denote the stage game at state $s$, where $\overline{Q}_i^t(s) = [\overline{Q}_i^t(s, a)]_{a \in \mathcal{A}}$ is the game matrix of player $i$ comprised of the $Q$-value estimates at time $t$. Recall that a stage game ARNEQ $(\overline{\pi}_i^t(\cdot|s), \widehat{\pi}^{i,t}(\cdot|s), \omega_i^t)_{i \in \mathcal{N}}$ for state $s$ is defined as the joint policy that satisfies, for each $i$, the following:

$$(\overline{\pi}_i^t(\cdot|s), \widehat{\pi}^{i,t}(\cdot|s), \omega_i^t) \in \arg \max_{\pi_i \in \Pi_i} \min_{\substack{\omega_i \in \Omega_i \\ \widehat{\pi}^i \in \Pi_{-i}}} \mathbb{E}_{K \sim \omega_i} \left[ \sum_{a \in \mathcal{A}} \pi_i(a_i|s) \prod_{i \neq j \notin K} \overline{\pi}_j^t(a_j|s) \prod_{l \in K} \widehat{\pi}_l^i(a_l|s) \overline{Q}_i^t(s, a) \right]$$

We restate Assumption 3 for convenience.

**Statement 2** *For each stage game at time $t$ and state $s \in \mathcal{S}$, one of the following conditions holds.*

- *A stage ARNEQ is also a global optimum, that is, for any $i \in \mathcal{N}$ and $\pi(\cdot|s)$, we have*

$$\mathbb{E}_{K \sim \omega_i^t} \left[ \sum_{a \in \mathcal{A}} \pi_{-K}^t(a_{-K}|s) \widehat{\pi}_K^{i,t}(a_K|s) \overline{Q}_i^t(s, a) \right] \geq \mathbb{E}_{K \sim \omega_i} \left[ \sum_{a \in \mathcal{A}} \pi_{-K}(a_{-K}|s) \widehat{\pi}_K^i(a_K|s) \overline{Q}_i^t(s, a) \right] \ ,$$

*for any $\pi \in \Pi$, $\widehat{\pi}^i \in \Pi_{-i}$ and $\omega_i \in \Omega_i$.*

- *A given player's payoff is increased if other benign players or its attacker deviate, that is, for any $i \in \mathcal{N}$ we have*

$$
\mathbb{E}_{K \sim \omega_i^t} \left[ \sum_{a \in \mathcal{A}} \pi_{-K}^t(a_{-K}|s) \widehat{\pi}_K^{i,t}(a_K|s) \overline{Q}_i^t(s,a) \right] \leq \mathbb{E}_{K \sim \omega_i'} \left[ \sum_{a \in \mathcal{A}} \pi_i^t(a_i|s) \pi_{-(K \cup \{i\})}'(a_{-(K \cup \{i\})}|s) \widehat{\pi}_K'(a_K|s) \overline{Q}_i^t(s,a) \right] ,
$$

*for all $\pi' \in \Pi$, $\widehat{\pi}' \in \Pi_{-i}$ and $\omega_i' \in \Omega_i$.*

Given that Assumptions 1, 2 and 3 hold, we can proceed with our argument. We will prove the following result.

**Lemma 8** *For every player $i$ and round $t$, and for any given stage game matrix of $Q$-values $\overline{Q}_i(s)$, let $\overline{\mathcal{B}}_i^t : \mathbb{Q} \to \mathbb{Q}$ be an operator on the $Q$-value function space $\mathbb{Q}$, defined as*

$$
\overline{\mathcal{B}}_i^t \overline{Q}_i(s,a) := \mathcal{R}_i(s,a) + \gamma \mathbb{E}_{K \sim \omega_i^t} \left[ \sum_{a' \in \mathcal{A}} \overline{\pi}_{-K}^t(a_{-K}'|s) \widehat{\pi}_K^{i,t}(a_K'|s) \overline{Q}_i(s,a') \right] ,
$$

*for any given $(s,a)$-tuple. Then, $\overline{\mathcal{B}}_i^t$ is a contraction operator with respect to the $l_\infty$ norm.*

In order to prove the above lemma, we will make use of the following standard result on pseudo-contractions.

**Lemma 9 (Lemma 8 of Hu & Wellman (2003))** *Assume that $\alpha_t$ satisfy Assumption 2 and that the mapping $\mathcal{B}^t : \mathbb{Q} \to \mathbb{Q}$ satisfies the following condition: there exists a number $0 < \gamma < 1$ and a sequence $\gamma_t \geq 0$ converging to zero with probability 1 such that*

$$
\left\| \mathcal{B}^t Q - \mathcal{B}^t Q^* \right\| \leq \gamma \left\| Q - Q^* \right\| + \gamma_t ,
$$

*for all $Q \in \mathbb{Q}$ and $Q^*$ such that $Q^* = \mathbb{E}[\mathcal{B}^t Q^*]$, then the iteration defined by*

$$
Q_{t+1} = (1 - \alpha_t) Q_t + \alpha_t \mathcal{B}^t Q_t
$$

*converges to $Q^*$ with probability 1.*

Now we are ready to prove Lemma 8. In order to do so, we need to show that its conditions are satisfied for our setting.

**Proof:** Let $(\overline{Q}_i)_{i \in \mathcal{N}}$ and $(\overline{Q}_i')_{i \in \mathcal{N}}$ be two given sets of $Q$-value functions. Denote by $(\overline{\pi}_i, \widehat{\pi}^i, \omega_i)_{i \in \mathcal{N}}$ and $(\overline{\pi}_i', \widehat{\pi}', \omega_i')_{i \in \mathcal{N}}$ their corresponding ARNEQ policies, respectively. Let $(s,a)$ be a given state-action tuple and suppose we have

$$
\overline{\mathcal{B}}_i^t \overline{Q}_i(s,a) \geq \overline{\mathcal{B}}_i^t \overline{Q}_i'(s,a).
$$

If the first part of Assumption 3 holds, then we have

$$
0 \leq \left| \overline{\mathcal{B}}_i^t \overline{Q}_i(s,a) - \overline{\mathcal{B}}_i^t \overline{Q}_i'(s,a) \right|
$$

$$
= \gamma \left| \mathbb{E}_{K \sim \omega_i} \left[ \sum_{a' \in \mathcal{A}} \overline{\pi}_{-K}(a_{-K}'|s) \widehat{\pi}_K^i(a_K'|s) \overline{Q}_i(s,a') \right] - \mathbb{E}_{K \sim \omega_i'} \left[ \sum_{a' \in \mathcal{A}} \overline{\pi}_{-K}'(a_{-K}'|s) \widehat{\pi}_K'(a_K'|s) \overline{Q}_i'(s,a') \right] \right|
$$

$$
\leq \gamma \left| \mathbb{E}_{K \sim \omega_i} \left[ \sum_{a' \in \mathcal{A}} \overline{\pi}_{-K}(a_{-K}'|s) \widehat{\pi}_K^i(a_K'|s) \overline{Q}_i(s,a') \right] - \mathbb{E}_{K \sim \omega_i} \left[ \sum_{a' \in \mathcal{A}} \overline{\pi}_{-K}(a_{-K}'|s) \widehat{\pi}_K^i(a_K'|s) \overline{Q}_i'(s,a') \right] \right|
$$

$$
= \gamma \left| \mathbb{E}_{K \sim \omega_i} \left[ \sum_{a' \in \mathcal{A}} \overline{\pi}_{-K}(a_{-K}'|s) \widehat{\pi}_K^i(a_K'|s) \left( \overline{Q}_i(s,a') - \overline{Q}_i'(s,a') \right) \right] \right|
$$

$$
\leq \gamma \max_{K \in \mathcal{N}_i} \left| \sum_{a' \in \mathcal{A}} \overline{\pi}_{-K}(a_{-K}'|s) \widehat{\pi}_K^i(a_K'|s) \left( \overline{Q}_i(s,a') - \overline{Q}_i'(s,a') \right) \right|
$$

$$\leq \gamma \max_{a' \in \mathcal{A}} \left| \overline{Q}_i(s, a') - \overline{Q}'_i(s, a') \right|$$

$$\leq \gamma \left\| \overline{Q}_i - \overline{Q}'_i \right\|_\infty .$$

If the second part of Assumption 3 holds, then, instead we have

$$0 \leq \left| \overline{\mathcal{B}}_i^t \overline{Q}_i(s, a) - \overline{\mathcal{B}}_i^t \overline{Q}'_i(s, a) \right|$$

$$= \gamma \left| \mathbb{E}_{K \sim \omega_i} \left[ \sum_{a' \in \mathcal{A}} \overline{\pi}_{-K}(a'_{-K}|s) \widehat{\pi}^i_K(a'_K|s) \overline{Q}_i(s, a') \right] - \mathbb{E}_{K \sim \omega'_i} \left[ \sum_{a' \in \mathcal{A}} \overline{\pi}'_{-K}(a'_{-K}|s) \widehat{\pi}'_K(a'_K|s) \overline{Q}'_i(s, a') \right] \right|$$

$$\leq \gamma \left| \mathbb{E}_{K \sim \omega_i} \left[ \sum_{a' \in \mathcal{A}} \overline{\pi}_{-K}(a'_{-K}|s) \widehat{\pi}^i_K(a'_K|s) \overline{Q}_i(s, a') \right] - \mathbb{E}_{K \sim \omega'_i} \left[ \sum_{a' \in \mathcal{A}} \overline{\pi}_i(a_i|s) \overline{\pi}'_{-K-i}(a'_{-K-i}|s) \widehat{\pi}'_K(a'_K|s) \overline{Q}_i(s, a') \right] \right|$$

$$\leq \gamma \left| \mathbb{E}_{K \sim \omega'_i} \left[ \sum_{a' \in \mathcal{A}} \overline{\pi}_i(a_i|s) \overline{\pi}'_{-K-i}(a'_{-K-i}|s) \widehat{\pi}'_K(a'_K|s) \overline{Q}_i(s, a') \right] \right.$$

$$\left. - \mathbb{E}_{K \sim \omega'_i} \left[ \sum_{a' \in \mathcal{A}} \overline{\pi}_i(a_i|s) \overline{\pi}'_{-K-i}(a'_{-K-i}|s) \widehat{\pi}'_K(a'_K|s) \overline{Q}'_i(s, a') \right] \right|$$

$$\leq \gamma \max_{K \in \mathcal{N}_i} \left| \sum_{a' \in \mathcal{A}} \overline{\pi}_i(a_i|s) \overline{\pi}'_{-K}(a'_{-K}|s) \widehat{\pi}'_K(a'_K|s) \left( \overline{Q}_i(s, a') - \overline{Q}'_i(s, a') \right) \right|$$

$$\leq \gamma \max_{a' \in \mathcal{A}} \left| \overline{Q}_i(s, a') - \overline{Q}'_i(s, a') \right|$$

$$\leq \gamma \left\| \overline{Q}_i - \overline{Q}'_i \right\|_\infty ,$$

where the second inequality follows from the equilibrium definition with respect to player $i$'s policy and the third inequality follows by the second part of Assumption 3.

A similar argument is used for the case when we have

$$\overline{\mathcal{B}}_i^t \overline{Q}_i(s, a) \leq \overline{\mathcal{B}}_i^t \overline{Q}'_i(s, a).$$

Since the chosen $(s, a)$-tuple was arbitrary, we conclude that

$$\left\| \overline{\mathcal{B}}_i^t \overline{Q}_i - \overline{\mathcal{B}}_i^t \overline{Q}'_i \right\|_\infty \leq \gamma \left\| \overline{Q}_i - \overline{Q}'_i \right\|_\infty .$$

$$\square$$

Thus, we have shown that the operator $\overline{\mathcal{B}}_i^t$ is a contraction. Recall that this operator has $\overline{Q}_i^*$ as a fixed point. Furthermore, the rates $\alpha_t$, $t \geq 1$, satisfy the requirements of Lemma 9. Hence, the result of Theorem 2 follows.

## C  Experiments: Environments

### C.1  Independent Spread

This environment modifies the Spread environment from Lowe et al. (2017) in such a way, that all agents pursue an individual goal, instead of cooperating toward one shared objective. Each agent strives to reduce the distance to a landmark that is designated to them while simultaneously avoiding collisions with other agents. Therefore, the reward of each agent consists of the negative $L_2$-distance to their designated landmark, coupled with a large penalty of 10 for colliding with agents. Agents observe their position, as well as the relative position of their designated landmark and all peer agents. The action space consists of no-action as well as increasing the velocity in one of four directions.

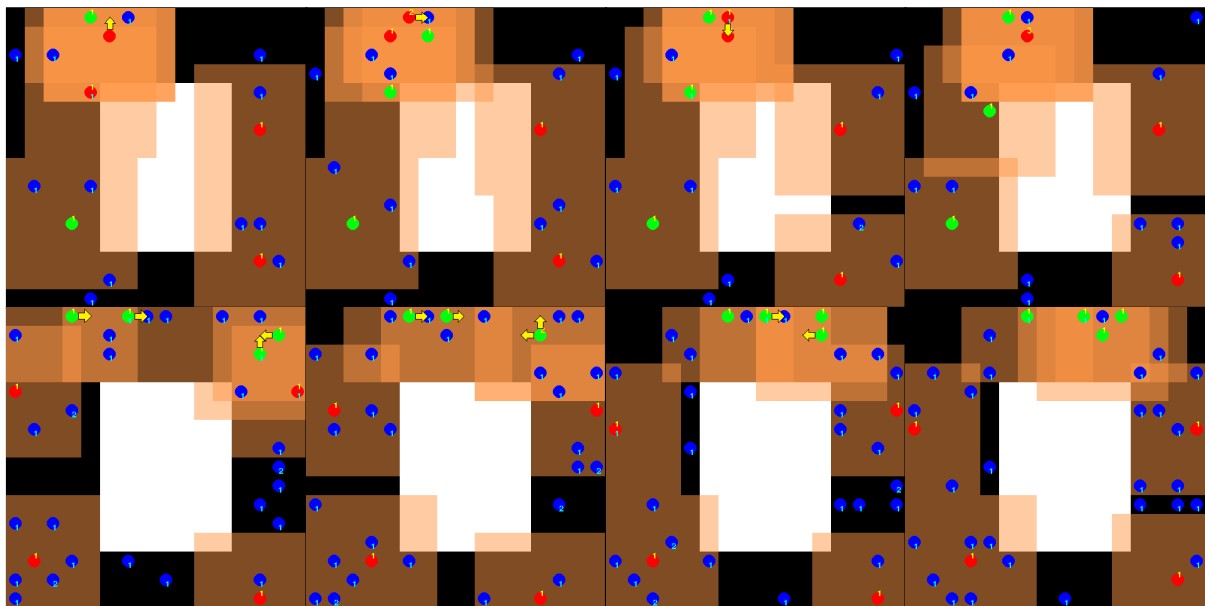

Figure 6: Full version of Figure 4 (zoomed out).

## C.2 Multi-Agent Ant

We consider a cooperative multi-agent extension of the MuJoCo Ant environment (Todorov et al., 2012) as proposed in Peng et al. (2021). Agents control one leg of the ant, and need to cooperate with the other legs to maximize the distance traveled forward, while maintaining balance. Consequentially, each agent receives a shared reward, which consists of the forward momentum compared to the preceding frame and a small incentive of 0.1 reward for not falling over. We removed the penalty for large actions and high external contact, as the adversary could easily exploit this reward function without actually impacting the ant's objective and there is defence mechanism for the other agents. Agents observe the position and orientation of all joints and can apply torque to the joint between the torso and upper leg and between the upper and lower leg of the leg they control.

## C.3 Pursuit

We finally consider the cooperative Pursuit environment Gupta et al. (2017). Each agent assumes the role of one of the eight pursuers aiming to catch 30 randomly moving evaders. Evaders are captured if they are enclosed on all sides by pursuers or walls. Pursuers receive the reward of 5 for every successful capture, as well as an urgency reward of $-0.1$ every time step there is still prey left to be captured. Observations consist of all pursuers, evaders, and walls within a 7x7 grid around the agent. The available actions are staying at the current position or moving one field to the left, right, up, or down.

# D Experiments: Hyperparameters

Hyperparameters for the PPO algorithm for all environments are given in Table 7.

| Hyperparameter | Value for Spread | Value for Ant | Value for Pursuit |
|---|---|---|---|
| Total timesteps | $800K$ | $13M$ | $10M$ |
| Batch size | 4000 | 65536 | 5000 |
| Minibatch size | 128 | 4096 | 500 |
| Epochs per update | 30 | 30 | 30 |
| Learning rate $\alpha$ | 0.0003 | $5 * 10^{-5}$ | $5 * 10^{-5}$ |
| Discount Factor $\gamma$ | 0.99 | 0.99 | 0.99 |
| Clipping parameter A | 0.3 | 0.2 | 0.3 |
| Advantage estimation discount $\lambda$ | 1.0 | 0.95 | 0.95 |
| Entropy coefficient | 0.0 | 0.0 | 0.01 |
| Value function loss coefficient | 1.0 | 0.5 | 1.0 |

Figure 7: Hyperparameters for Spread, Ant, and Pursuit

## E   Experiments: Computing Infrastructure

Computing infrastructure for the Independent Spread and Pursuit environment:

- **GPU:** None

- **CPU:** Intel Xeon E5-2667 v2

- **Memory:** 256GB, DDR3, 1866 MHz, ECC

- **Operating System:** Debian

The computing infrastructure for the Ant environment:

- **GPU:** V100 Nvidia Tesla GPU 32GB

- **CPU:** Intel Xeon Gold 6134M

- **Memory:** 768GB, DDR4 2666MT/s, ECC

- **Operating System:** Debian

## F   Experiments: Runtime

Runtime for all algorithms and environments are given in Table 8.

| Algorithm | Runtime for Spread | Runtime for Ant | Runtime for Pursuit |
|---|---|---|---|
| Independent Learning | 27m28s | 1h45m58s | 10h6m |
| fixed-k | 22m36s | 1h46m48s | 10h9m |
| ART | 34m38s | 1h50m24s | 10h55m59s |

Figure 8: Runtime of algorithms for Spread, Ant and Pursuit

