# OpenReview forum: "Defending Against Unknown Corrupted Agents: Reinforcement Learning of Adversarially Robust Nash Equilibria"
_TMLR — Accepted by TMLR_

### Review · Reviewer_hWsb · 2024-05-23

**Summary Of Contributions:**

In this paper, the authors introduce ARNEQ, a equilibrium concept for multiplayer games, where each agent i is robust to adversarial opponent strategies that are trying to reduce i's utility rather than maximizing their own utility. The authors provide an argument for existence, and an impractical algorithm for finding it. Inspired by those two things, the authors also provide a practical NN-based algorithm, where an ARNEQ would be a globally optimal solution. The paper has experiments on a few domains using the ARNEQ adversary attacks, showing the algorithm produces a policy with higher value under attack than a baseline of independent PPO.

**Audience:**

Yes

**Claims And Evidence:**

Yes

**Requested Changes:**

“In our setting, this information is not given. As a consequence, there is no fixed game to be solved and thus, no well-defined NE.”
Possibly just say something like “As a consequence, the problem is no longer a Markov game, and the solution is not a NE in that space.”  It is no longer a Markov game, but it’s not so obvious there’s no fixed game within a larger class of games, especially if you break the assumption that all states are reachable in a single episode: e.g., something like a POMDP that uniformly mixes over starting states that determine which agent the adversary is targeting, then let the adversary choose what agents to replace, then follow the original game dynamics except the controlled agents actions are ignored in favor of the adversary actions.

Section 5.2 Figure 5: This measurement seems like it's measuring something slightly different than ARNEQ convergence: once the adversary is fixed, the player is free to drift further away from a robust policy because the adversary is no longer tracking and responding to that new behavior. The non-trivial decrease when all other agents and the adversary are fixed is puzzling, and possibly mildly concerning.
Leaving aside the decrease in value after further traning, I think it’s reasonable to say low values in Figure 5 do indicate convergence as they’re upper bounds – the true increase in ARNEQ value for an agent should be less than leaving the adversary fixed, because the true value is a minimum over all adversary choices. I also think that extra step of indirection should be stated.


------ Some smaller issues for clarity ------

“this formulation renders the classical NE notion obsolete”  obsolete -> inappropriate?

Section 3.2: V bar, Q bar*, ARNEQ: these all have very similar terms in the sum, but are all written slightly differently. It seems worth the effort to have a more consistent choice throughout, rather than shuffling terms and grouping them differently.

For Q bar* and ARNEQ, and the recursive update for ARNEQ after equation 2, I found the removal of min omega, pi hat from the equation and treating them as given to be less clear.

Section 4, adversary loss on omega: add a general description in plain language. Something like "the loss for the adversary to agent i is agent i's ARNEQ value"?

Section 4, adversary loss on pi hat: if it requires three lines to explain the difference from the loss on omega, the actual equation should also be included

Section 5.2 “Agents trained with ART learned to meet at a specific rendezvous point unknown to the adversary”   Unknown because the adversary was not fully trained? Shouldn't the adversary be able to work out that rendezvous point, given access to (data describing) the agent policies?

**Strengths And Weaknesses:**

Producing tractable and useful equilibrium concepts for multiplayer environments is difficult, and robustness to unexpected interest is of interest a broad community, so a robust multiplayer equilibrium concept for Markov games seems like a contribution that could be of interest to a range of readers. The NN-based algorithm does not introduce a lot of extra complication on top of an existing multiplayer algorithm, making it worth trying in situations where robustness might be a concern.

One thing that might be missing is an argument for the form of attack being reasonable -- plausible use cases where this would be the right kind of attack to expect. As noted in the experiments looking at performance with no attack, there is a cost to defending against an attack that might not happen, and a not quite paranoid assumption that "some number of agents might all collude to maximally impact my value with no respect to their own utility" could be quite pessimistic in some environments.

---

> ### Author Response · Authors · 2024-06-28
>
> We thank the reviewer for the helpful comments. Below, we have tried to address all of them.
>
> **_“In our setting, this information is not given. As a consequence, there is no fixed game to be solved and thus, no well-defined NE.” Possibly just say something like “As a consequence, the problem is no longer a Markov game, and the solution is not an NE in that space.” It is no longer a Markov game, but it’s not so obvious there’s no fixed game within a larger class of games, especially if you break the assumption that all states are reachable in a single episode: e.g., something like a POMDP that uniformly mixes over starting states that determine which agent the adversary is targeting, then let the adversary choose what agents to replace, then follow the original game dynamics except the controlled agents’ actions are ignored in favor of the adversary actions._**
>
> We understand the reviewer’s concern with the current phrasing. We have changed the phrasing as the reviewer suggested in the revised version of the paper.
>
> **_Section 5.2 Figure 5: This measurement seems like it's measuring something slightly different than ARNEQ convergence: once the adversary is fixed, the player is free to drift further away from a robust policy because the adversary is no longer tracking and responding to that new behavior. The non-trivial decrease when all other agents and the adversary are fixed is puzzling, and possibly mildly concerning. Leaving aside the decrease in value after further traning, I think it’s reasonable to say low values in Figure 5 do indicate convergence as they’re upper bounds – the true increase in ARNEQ value for an agent should be less than leaving the adversary fixed, because the true value is a minimum over all adversary choices. I also think that extra step of indirection should be stated._**
>
> We thank the reviewer for bringing up this point. We would like to note that the definition of an equilibrium implies that no player can be better by deviating from its equilibrium strategy when all other players are fixed. It also implies that if an agent deviates from an ARNEQ, this could only be beneficial for its potential adversaries. And that is precisely what we are trying to measure in Figure 5. If indeed all players have converged to an ARNEQ, then, further training a given agent when the rest of the players keep playing fixed strategies, should not imply increased performance for this player. Otherwise, this would mean there is room for improvement which would violate the very condition of an equilibrium. Thus, by showing that there is, on average, negligible improvement for a given agent, we are effectively showing that players converge to an approximate ARNEQ. Regarding the second point, the decrease in performance is related to oscillations in training curves which come from the fact that the learning procedure is not stable in such environments. However, on average we are still able to show that the change in performance is negligible.
>
> **_“this formulation renders the classical NE notion obsolete” obsolete -> inappropriate?_**
>
> The suggested change is reflected in the revised version of the paper.
>
> **_Section 3.2: V bar, Q bar*, ARNEQ: these all have very similar terms in the sum, but are all written slightly differently. It seems worth the effort to have a more consistent choice throughout, rather than shuffling terms and grouping them differently._**
>
> We agree with the reviewer that the notation used throughout those definitions may seem a bit redundant, since they do indeed contain similar terms. Therefore, we have modified the definition of \bar V to maintain the same style as in the definitions of \bar Q^* and ARNEQ in Section 3.2
>
> **_For Q bar* and ARNEQ, and the recursive update for ARNEQ after equation 2, I found the removal of min omega, pi hat from the equation and treating them as given to be less clear._**
>
> Since \bar Q^* denotes an equilibrium, and since this equilibrium is defined in terms of benign policies \pi, adversarial policies \hat \pi, and subset selection policies \omega, there is no further need to minimize over one of these when at an equilibrium. Note that, at the end, we do not need to know the values of \hat \pi and \omega, since we only care about the benign policies. Nevertheless, these values exist at an equilibrium which is what we denote by their equilibrium counterparts.

---

> > ### Comment · Reviewer_hWsb · 2024-07-02
> >
> > Thanks to the authors for their response and changes addressing most of the questions and concerns I had. I do still think there should be an additional revision -- I don't think it needs a lot of text, but I do think the issue should be resolved.
> >
> >
> > **Section 5.2 Figure 5: This measurement seems like it's measuring something slightly different than ARNEQ convergence: once the adversary is fixed, the player is free to drift further away from a robust policy because the adversary is no longer tracking and responding to that new behavior. The non-trivial decrease when all other agents and the adversary are fixed is puzzling, and possibly mildly concerning. Leaving aside the decrease in value after further traning, I think it’s reasonable to say low values in Figure 5 do indicate convergence as they’re upper bounds – the true increase in ARNEQ value for an agent should be less than leaving the adversary fixed, because the true value is a minimum over all adversary choices. I also think that extra step of indirection should be stated.**
> >
> > > We thank the reviewer for bringing up this point. We would like to note that the definition of an equilibrium implies that no player can be better by deviating from its equilibrium strategy when all other players are fixed. It also implies that if an agent deviates from an ARNEQ, this could only be beneficial for its potential adversaries. And that is precisely what we are trying to measure in Figure 5. If indeed all players have converged to an ARNEQ, then, further training a given agent when the rest of the players keep playing fixed strategies, should not imply increased performance for this player. Otherwise, this would mean there is room for improvement which would violate the very condition of an equilibrium. Thus, by showing that there is, on average, negligible improvement for a given agent, we are effectively showing that players converge to an approximate ARNEQ. Regarding the second point, the decrease in performance is related to oscillations in training curves which come from the fact that the learning procedure is not stable in such environments. However, on average we are still able to show that the change in performance is negligible.
> >
> > I have no issues with the underlying argument for figure 5: as I note in the original comment, I think the figure *is* reasonable indirect evidence for being close to an eq'm. My issue is with the fact that the argument is indirect, and the paper not making that indirect argument clear within the text. Questions about this argument also seems to be raised by reviewer aM2t. Condense the arguments in the author responses and include it in the paper: make the argument clear there.
> >
> > Regarding the decrease in performance, I would also suggest -- not as strongly as the point above, but I would still suggest it -- that the authors give some context for the magnitude of values, or the variance in learning. I'm willing to believe that the issue is just noise in learning curves as the authors suggest, but "it sounds plausible" is not a well-evidence claim.

---

> > > ### Author Response · Authors · 2024-07-04
> > >
> > > We thank the reviewer for the additional feedback on the submission. To address the reviewer's comments related to the explanation and decrease in value, we have decided to add the following paragraph as additional text in the caption of Figure 5:
> > >
> > > "If the players have converged to an ARNEQ, then further training a given agent when the rest of the players keep playing fixed strategies should not imply increased performance for this player. Otherwise, this would mean there is room for improvement which would violate the very condition of an equilibrium. Note that, on average, performance tends to stay the same, except for some small oscillations which are due to the stochasticity of deep RL methods. These oscillations also align with the standard error depicted in Table 1, so they are within the expected variance in rewards. This suggests that players are close to an ARNEQ."
> > >
> > > We hope this addresses the reviewer's concerns.

---

> ### Author Response · Authors · 2024-06-28
>
> **_Section 4, adversary loss on omega: add a general description in plain language. Something like "the loss for the adversary to agent i is agent i's ARNEQ value"?_**
>
> We have rephrased the explanation of the definition based on the reviewer’s suggestion.
>
> **_Section 4, adversary loss on pi hat: if it requires three lines to explain the difference from the loss on omega, the actual equation should also be included_**
>
> The loss equation is exactly the same as the previous loss, only with respect to \hat \pi. That is why we have omitted repeating the same quantity.
>
> **_Section 5.2 “Agents trained with ART learned to meet at a specific rendezvous point unknown to the adversary” Unknown because the adversary was not fully trained? Shouldn't the adversary be able to work out that rendezvous point, given access to (data describing) the agent policies?_**
>
> We have used the phrasing “unknown to the adversary” in the sense that the benign agents have learned how to effectively avoid contact with other agents, since they cannot distinguish between adversary and benign agent. Instead, they choose to cooperatively surround the evaders at specific locations. This technique ensures that in every group of agents, there are always enough benign agents present to ensure that the adversary cannot intervene in the process of surrounding prey.

---

### Review · Reviewer_MAs3 · 2024-05-26

**Summary Of Contributions:**

The paper considers a multiagent reinforcement learning (MARL) setting in which an adversary can arbitrarily corrupt a subset of the agents' policies at test time. To tackle such cases, the authors introduce a new solution concept, called adversarially robust Nash equilibrium (ARNEQ). They prove that ARNEQ always exists, and they provide a model-based algorithm that computes an ARNEQ. Moreover, in light of the impracticality of the previous algorithm, they also propose a more practical independent learning algorithm based on stochastic gradient descent, albeit with no theoretical guarantees. Experiments on standard benchmarks demonstrate the value of the proposed method in addressing adversarial corruptions in MARL.

**Audience:**

Yes

**Broader Impact Concerns:**

I do not have any concerns on the ethical implications of the work.

**Claims And Evidence:**

Yes

**Requested Changes:**

I have the following comments, none of which is critical for securing acceptance.

- The citation style is often used incorrectly. For example, in Section 1.1.1, it should read "Adversarial attacks on RL systems have been extensively studied in recent years (Kiourti et al. (2020))" instead of "Adversarial attacks on RL systems have been extensively studied in recent years Kiourti et al. (2020)..." Namely, use parentheses when it is not part of the sentence.
- I am a bit confused as to the interpretation of Assumption 3, which does not seem standard. It would be nice if the authors can elaborate more on that. Also, the first bullet in Assumption 3 has to be explained further in the main body as it is not clear at all what it means.
- There is a missing period (“.”) at the last displayed equation in page 23.

**Strengths And Weaknesses:**

Strengths: The paper considers a very natural and well-motivated problem, as described above. Variants of that problem have already received attention in prior research, but to the best of my knowledge the problem has not been treated in this generality before. The paper does a great job at discussing related work, and I did not find any notable omissions. The story and the results of the paper are quite complete and compelling, and take an important step towards building tools to tackle adversarial corruptions in MARL. All results appear to be sound, and I did not detect any notable issues. The paper is also very well written and organized. I overall believe that the paper makes a concrete contribution on an important problem, and therefore deserves to be accepted for presentation.

Weaknesses: On the negative side, the paper has arguably a limited technical contribution, and although the results are new, it heavily relies on known techniques and results. Moreover, the main algorithm used in the experiments does not have theoretical guarantees of convergence; nonetheless, this is to be expected as even computing Nash equilibria is known to be a hard problem. On that note, one suggestion for future work would be to follow the approach of the recent paper "Approximating Nash Equilibria in Normal-Form Games via Unbiased Stochastic Optimization" (presented at ICLR 2024). Finally, it would be interesting to see more experiments in the regime where k and n are much larger, but I understand that this could require different techniques and it is left for future work.

---

> ### Author Response · Authors · 2024-06-28
>
> We thank the reviewer for the helpful comments. Below, we have tried to address all of them.
>
> **_The citation style is often used incorrectly. For example, in Section 1.1.1, it should read "Adversarial attacks on RL systems have been extensively studied in recent years (Kiourti et al. (2020))" instead of "Adversarial attacks on RL systems have been extensively studied in recent years Kiourti et al. (2020)..." Namely, use parentheses when it is not part of the sentence._**
>
> We agree with the reviewer that the style of the citations is often not the correct one, and we have fixed all citation styles throughout the revised version of the paper, based on the reviewer’s suggestion.
>
> **_I am a bit confused as to the interpretation of Assumption 3, which does not seem standard. It would be nice if the authors can elaborate more on that. Also, the first bullet in Assumption 3 has to be explained further in the main body as it is not clear at all what it means._**
>
> We understand the reviewer’s confusion related to Assumption 3. We would like to note that Assumption 3 has been previously used in the MARL literature for theoretical analysis of centralized learning methods – Assumption 3 in (Yang et al., 2018), Assumption 3 in (Hu & Wellman, 2003) and Assumption 4.3 in (Zhang et al., 2020). As such, it is standard in this context. The exact formulation of Assumption 3 was previously given in Statement 2 of Appendix B. We have also added it in full in the main paper for clarity. As noted in (Hu & Wellman, 2003), this assumption, although restrictive, is not needed in practice. However, it is essential to theoretically show convergence. We have also added a proof sketch to further provide intuition on how all assumptions are used in the proof of the result. We reproduce the proof sketch here for convenience.
>
> Proof sketch for Theorem 2. The main ingredient of the proof is utilizing a previous result from (Hu & Wellman, 2003) which states that, if Assumptions 1 and 2 hold, and a given operator on the Q-functions is a contraction, then the procedure described above converges to an equilibrium. Hence, it suffices to prove that the ARNEQ operator is a contraction. We use Assumption 3 to that end, by separately considering both cases of the assumption. With this, all the conditions of the utilized result are satisfied, and thus we conclude convergence to an ARNEQ.
>
> **_There is a missing period (“.”) at the last displayed equation in page 23._**
> We thank the reviewer for pointing out the typo. We have fixed it in the revised version of the paper.

---

> > ### Comment · Reviewer_MAs3 · 2024-07-15
> >
> > I thank the authors for the detailed response. My comments have been adequately addressed.

---

### Review · Reviewer_aM2t · 2024-06-18

**Summary Of Contributions:**

The authors propose an interesting "Adversarially Robust Nash Equilibrium" (ARNEQ). They describe a realistic setting for adversarial attacks in Multi-Agent Reinforcement Learning and state how the ARNEQ is a useful notion of robustness in this setting. They position their work well with related work and their contributions are clearly stated. They provide an extensive proof of existence for ARNEQs in their specific setting. Furthermore, they present a model-based algorithm which provably finds ARNEQs. Next, they provide a model-free algorithm that intuitively should approximate ARNEQs. The authors demonstrate that the model-free method learns distinct policies from their reasonable baselines in three distinct, sufficiently high-dimensional environments. The ARNEQ and the model-free method are well motivated and simple, which adds to their potential impact on and relevancy for the field of adversarial attacks in Multi-Agent RL.

**Audience:**

Yes

**Broader Impact Concerns:**

I do not think that the current work requires a broader impact statement

**Claims And Evidence:**

Yes

**Requested Changes:**

Questions and Requested Changes (RC):
- *RC1 for proofs*: The proofs could benefit for more specific motivations at each step. Ideally, the authors would point out, when reasonable, if there are any relevant consequences for any assumption they make. For example, why does a correspondence need to be semicontinuous? The recommendation would not change, as the proofs, if correct, seem sufficiently detailed for a more knowledgable audience. Providing more details would simply make the proofs more understandable for a less knowledgable audience, which would also facilitate future work.

- *RC2: For Section 4*: I believe it would be very helpful if the authors could clarify the consequences of using a model-free method as opposed to their previous model-based method. At the moment it is unclear if any of the convergences transfer or if any other connection can be made between the method in Section 3 and Section 4. They appear completely disconnected except for the intuitively similar objective functions. I believe the connection between Section 3 and 4 could be strengthened with different changes. First, the authors could simply state that any previous convergence guarantees do not hold anymore and it's not given that this method approaches an ARNEQ. Second, they could show that, in theory, ART still approximately converges to an ARNEQ (which they later claim empirically in Section 5.3.Convergence). Third, they could implement the method in Section 3 in a simpler environment, where it's computationally feasible to compute the model-based method, and provide empirical evidence that ART converges to the same equilibrium. Such a clarification would have an affect on securing a recommendation for acceptance.
- *Question for Algorithm 1*: Why do we optimize wrt $\theta^t_i$ and not $\theta^(t-1)_i$ in Line 4 and 6? In a similar vein, if we already have a model of the opponent, why shouldn't we update our parameters $\bar{\theta}^{t-1}$ with respect to the parameter that the opponent will have in the next step $\widehat{\theta}_j^{i, t}$? Why would I update with respect to $t-1$?  For example, first you could update $\widehat{\theta}_j^{i, t}\leftarrow \mathcal{L}(\theta_i^t, \bar{\theta}^{t-1}, \widehat{\theta}^{i, t-1})$ and then $\bar{\theta}^{t} \leftarrow \mathcal{L}(\theta_i^t, \bar{\theta}^{t-1}, \widehat{\theta}^{i, t})$. That way you can approximate a best response to the update of the opponent will have. See Opponent Shaping research, especially LOLA [1] and LOQA [2], for related work. While I do not think the current version is incorrect, I think the authors could better motivate why we optimize with respect to $\theta^t_i$ but not with respect to $\widehat{\theta}_j^{i, t}$, which would improve the paper.
- *Question for Table 1*:   Why is (Fixed-k=2) == (ART-k=2). Doesn't (Fixed-k=2) always train against two adversaries and (ART-k=2) samples adversaries between $[1,2]$ as $\Omega_i=\Delta\left(\mathcal{N}_i\right)$ and $\mathcal{N_i}$ is the set of subsets of cardinality no greater than $k$. An answer to this question would probably also clarify some confusion for later questions and would affect in securing the recommendation.
- *Question for Section  5.3.Quantitative Analysis*: In the current version, the explanation for Pursuit is unsatisfying, though sufficient for acceptance. If it was hardness of training, wouldn't you expect Fixed-k=1 to perform better in the 4 adversary case than Fixed-k=2 or Fixed-k=4? It seems that the argument "hardness of training" could always apply and is thus not particularly insightful. Could you argue that k=1 might be the optimal adversarial policy $\theta$? But shouldn't $\theta$ in ART-k=4 learn that? In general, given that there are many pursuers agents, it is unintuitive why there is a discrepancy between Fixed-k=1 and ART-k=1. Maybe the authors could clarify?
- *Question for Section 5.3.Quantitative Analysis*: When the authors state "In Independent Spread, optimal performance of ART is achieved if the number of adversaries during training matches their number at deployment time", do they mean "maximum number of adversaries during training"?, since, to the best of my understanding, the number of actual adversaries is determined by $\theta$ and changes across training steps in ART-k.
- *Question for Section 5.3.Convergence*: Could the authors clarify what they mean by "A small increase signifies the approximate convergence of ART to an ARNEQ."? First, it is unclear why ART would converge even approximately to an ARNEQ, as this hasn't been shown theoretically to the best of my understanding. Second, could the authors clarify what they mean by "Depicted is the difference between the performance of the improved policy and the original performance." When they say "original performance", do they mean the performance of the originally benign-trained victim policy against the other benign policies or the benign victim policy against the adversarially trained policies? If it's the former, couldn't you argue that even a decrease in performance signifies converges to an ARNEQ?
- *RC3 for Section 5.3.Convergence*: In general, it feels insufficient to say that this experimental setup has anything to do with the ARNEQ. The authors could clarify how this is connected to the ARNEQ more specifically. Otherwise, to strengthen the claim that ART converges to an ARNEQ, it would be much more convincing if you had an environment where the method from Section 3 is feasible to compute and compare ART with the method from Section 3, which would greatly improve the papers results and improve the connection between Section 3 and Section 4. To repeat, I find it sufficient to say "we introduce a model-free gradient-based algorithm
that is able to empirically provide an efficient defense in various MARL environments.", which the authors do in Section 3. I find the evidence insufficient to say that "As the achieved improvement is small, this suggests that we approach an ARNEQ.". Even if ART does not approximate ARNEQs, I find the algorithm, empirical evaluation and general motivation interesting for the TMLR audience.

[1] Foerster, Jakob N., et al. "Learning with opponent-learning awareness." arXiv preprint arXiv:1709.04326 (2017).
[2] Aghajohari, Milad, et al. "LOQA: Learning with Opponent Q-Learning Awareness." arXiv preprint arXiv:2405.01035 (2024).

**Strengths And Weaknesses:**

I will keep strength and weaknesses concise and simply provide an overview. Please find more detailed questions in the requested changes.

Strengths:
First, the writing quality is good and the paper reads nicely. The notation style  is intuitive and the pseudocode provided facilitates understanding. Second, the author provide rigorous proofs for their theoretical contributions. They provide a proof of existence for their Adversarially Robust Nash Equilibrium (ARNEQ) under reasonable conditions. They also provide an interesting model-based method and a proof of convergence towards an ARNEQ for the model-based method. While the reviewer did not find any glaring mistakes in the proof, due to limited experience with such proofs, the reviewer cannot confirm with high certainty that there wouldn't be any errors. Questions regarding the proofs will be laid out in the requested changes.
A third strength is the proposal of a model-free method called Adversarially Robust Training (ART), that is algorithmically more convenient to compute in contrast to their proposed model-based method. The authors provide sufficient empirical evidence that ART learns policies distinct from Naive Learning that improve performance over the baseline statistically significantly over 5 seeds. The submission is relevant to TMLR and most claims are clear, convincing and accurate with a few exceptions.

Weaknesses:
The following section is meant to summarize where the reviewer thinks the paper could be improved the most. Detailed Questions and requested changes will be laid out in Requested Changes.

 First and foremost, the connection between the ARNEQ and ART is not sufficiently motivated. The transition from Section 3 to Section 4 simply states that the proposed model-based method is computationally infeasible and then a model-free method is proposed. While the objective function make intuitive sense, this transition could be improved through more theoretical or empirical results, or simply more clarification in writing. Second, and less importantly, some steps in the proofs of existence are hard to follow for an audience that is less experienced with such proofs. For example, can you motivate why you use Kakutani's fixed point theorem? Kakutani's fixed point theorem requires the definition of a semicontinuous correspondence. Does this assumption exclude any reasonable settings that the reader should be aware of? etc.

Third, Section 4 and 5 could benefit from some clarifications, which I will lay out later.

---

> ### Author Response · Authors · 2024-06-28
>
> We thank the reviewer for the helpful and insightful comments. Below, we have tried to address all of them.
>
> **_First and foremost, the connection between the ARNEQ and ART is not sufficiently motivated. The transition from Section 3 to Section 4 simply states that the proposed model-based method is computationally infeasible and then a model-free method is proposed. While the objective function make intuitive sense, this transition could be improved through more theoretical or empirical results, or simply more clarification in writing._**
>
> While we agree with the reviewer that there is indeed a gap between the theoretical and experimental parts of the paper, we would also like to point out that this structure is consistent with that of similar prior works (Zhang et al., 2020), where centralized methods are only introduced as a “proof of concept” with strong theoretical guarantees, and their decentralized counterparts are introduced as practical algorithms that can be readily applied.
>
> That said, we understand that the current transition may benefit from further elaboration. Thus, we have added a clarifying paragraph at the end of Section 3, as per the reviewer’s suggestion, which we hope will address the reviewer’s comment. We restate it here for convenience.
>
> “Although the proposed procedure is simple and intuitive, with the crucial benefit of satisfying strong theoretical guarantees, there are also several drawbacks associated with it. First, note that the update rule given in Equation (2) requires knowledge of the equilibrium policies of the benign agents for the stage games in every iteration, which in turn requires knowledge of the Q-values of all agents, from every agent’s point of view. This is a downside that all centralized, value-based, algorithms in MARL, such as Nash Q-Learning, share. Second, even if knowledge of the Q-values of all agents can be guaranteed, the problem of computing a Nash equilibrium from given utilities in a general-sum Markov game is known to be computationally hard (Daskalakis et al., 2009). Finally, note that the theoretical guarantees of the proposed method depend on the stated assumptions. Such assumptions may not always be satisfied in practice, where the irregularities in the individual utilities do not need to satisfy saddle point or global optima conditions. Motivated by the above, our next goal is thus to find a more practical and efficient approach to finding ARNEQ policies. In the next section, we introduce a model-free gradient-based algorithm that is able to empirically provide an efficient defense in various MARL environments.”
>
> **_Second, and less importantly, some steps in the proofs of existence are hard to follow for an audience that is less experienced with such proofs. For example, can you motivate why you use Kakutani's fixed point theorem? Kakutani's fixed point theorem requires the definition of a semicontinuous correspondence. Does this assumption exclude any reasonable settings that the reader should be aware of? Etc._**
>
> Kakutani’s fixed point Theorem is a standard result applied in proving the existence of any sort of fixed-point convergence of a set function. This makes it highly suitable for proving the existence of Nash equilibria in any type of game. The standard approach is to first define the best response correspondence when changing an action with every other action fixed, from each player’s point of view, and then show that such a function satisfies all the conditions of Kakutani’s theorem. It is worth pointing out that this approach is also traditionally used in the MARL literature (Hu & Wellman, 2003; Zhang et al., 2020; Kardeș et al, 2011).

---

> ### Author Response · Authors · 2024-06-28
>
> **_RC1 for proofs: The proofs could benefit for more specific motivations at each step. Ideally, the authors would point out, when reasonable, if there are any relevant consequences for any assumption they make. For example, why does a correspondence need to be semicontinuous? The recommendation would not change, as the proofs, if correct, seem sufficiently detailed for a more knowledgable audience. Providing more details would simply make the proofs more understandable for a less knowledgable audience, which would also facilitate future work._**
>
> It is worth pointing out that semicontinuity is not an assumption made on the function, but a condition of Kakutani’s fixed point theorem, which we show to be satisfied by our function defined at the beginning of Section A.1. Furthermore, we have added some further intuitive explanations in the Appendix related to the approach followed in the existence proof at the beginning of the Appendix. We have also added several other intuitive explanations, summaries, and descriptions of the ideas of the proofs in other places in the Appendix (all in blue). We reproduce the first paragraph here for convenience.
>
> For the existence proof, we will use the Kakutani's fixed point Theorem. This result has been classically used in proving the existence of Nash equilibria in various types of Markov games. It requires the careful construction of a set-valued function whose fixed point would represent the equilibrium of the game of interest. Once the construction is made, the theorem states that, if such a correspondence satisfies some technical conditions, then the existence of its fixed point is guaranteed, thus effectively proving the existence of an equilibrium of the game. This will be our approach in the following. First, we will prove some auxiliary results related to properties such as contraction, continuity, and convexity. Then, we will construct a set-valued function whose fixed point would represent an ARNEQ and further show that it satisfies the technical conditions of Kakutani's fixed point theorem.
>
> **_RC2: For Section 4: I believe it would be very helpful if the authors could clarify the consequences of using a model-free method as opposed to their previous model-based method. At the moment it is unclear if any of the convergences transfer or if any other connection can be made between the method in Section 3 and Section 4. They appear completely disconnected except for the intuitively similar objective functions. I believe the connection between Sections 3 and 4 could be strengthened with different changes. First, the authors could simply state that any previous convergence guarantees do not hold anymore and it's not given that this method approaches an ARNEQ. Second, they could show that, in theory, ART still approximately converges to an ARNEQ (which they later claim empirically in Section 5.3.Convergence). Third, they could implement the method in Section 3 in a simpler environment, where it's computationally feasible to compute the model-based method and provide empirical evidence that ART converges to the same equilibrium. Such a clarification would have an effect on securing a recommendation for acceptance._**
>
> As previously stated, we agree with the reviewer that there is indeed a discrepancy between the sections. We have tried to further smoothen the transition with the additional paragraph added in the first comment. It is important to point out that showing convergence guarantees for ART is not a feasible approach as independent learning is shown to be computationally intractable in general-sum Markov games. Thus, a theoretical analysis of ART is simply not possible.

---

> ### Author Response · Authors · 2024-06-28
>
> **_Question for Algorithm 1: Why do we optimize wrt $\theta^t_i$ and not $\theta^(t-1)_i$ in Line 4 and 6? In a similar vein, if we already have a model of the opponent, why shouldn't we update our parameters $\bar{\theta}^{t-1}$ with respect to the parameter that the opponent will have in the next step $\widehat{\theta}_j^{i, t}$? Why would I update with respect to $t-1$? For example, first you could update $\widehat{\theta}_j^{i, t}\leftarrow \mathcal{L}(\theta_i^t, \bar{\theta}^{t-1}, \widehat{\theta}^{i, t-1})$ and then $\bar{\theta}^{t} \leftarrow \mathcal{L}(\theta_i^t, \bar{\theta}^{t-1}, \widehat{\theta}^{i, t})$. That way you can approximate a best response to the update of the opponent will have. See Opponent Shaping research, especially LOLA [1] and LOQA [2], for related work. While I do not think the current version is incorrect, I think the authors could better motivate why we optimize with respect to $\theta^t_i$ but not with respect to $\widehat{\theta}_j^{i, t}$, which would improve the paper._**
>
> First, recall that $\theta^i_t$ denotes the adversarial subset selection policy maintained by player $i$. We first update this parameter in Line 3 so that player $i$ can subsequently sample the identities of its adversarial agents at time $t$. Once this is done, the rest of the updates regarding the benign policy of player $i$ and its adversarial players at that iteration are all done with respect to their previous counterparts in time $t-1$ in Lines 4 and 6. The dependence of the gradient on $\theta^i_t$ simply represents the fact that $\theta^i_t$ is used at the beginning of the round $t$ to sample the current adversaries (which are constantly changing and are only potential future adversaries, not real ones) to player $i$. Now, regarding why we update the benign policy first and adversaries next, we should clarify that, in our experiments, the same trajectories sampled with respect to $\theta^{t-1}$ and $\widehat{\theta}^{t-1}$ are used to update both benign and adversarial policies, so the order does not matter. This is done to reduce the sample complexity of the algorithm. We also noted that first updating the benign policies, then sampling more trajectories using the updated benign policies, and then using these trajectories to update the adversarial policies did not change the performance of the learners. As for the other way around (which is what the reviewer suggested) we have not tried that, but seems like an interesting approach for future direction.
>
> **_Question for Table 1: Why is (Fixed-k=2) == (ART-k=2). Doesn't (Fixed-k=2) always train against two adversaries and (ART-k=2) samples adversaries between $[1,2]$ as $\Omega_i=\Delta\left(\mathcal{N}_i\right)$ and $\mathcal{N_i}$ is the set of subsets of cardinality no greater than $k$. An answer to this question would probably also clarify some confusion for later questions and would affect in securing the recommendation._**
>
> The reviewer is right in pointing out that ART can also sample one instead of two adversaries in Independent Spread. However, note that, since the damage that two adversaries can do is always greater than the damage one adversary can do, it is obvious that the subset selection strategy will always prefer selecting two agents instead of one. Therefore, there is no essential distinction between the two.
>
> **_Question for Section 5.3.Quantitative Analysis: In the current version, the explanation for Pursuit is unsatisfying, though sufficient for acceptance. If it was hardness of training, wouldn't you expect Fixed-k=1 to perform better in the 4 adversary case than Fixed-k=2 or Fixed-k=4? It seems that the argument "hardness of training" could always apply and is thus not particularly insightful. Could you argue that k=1 might be the optimal adversarial policy $\theta$? But shouldn't $\theta$ in ART-k=4 learn that? In general, given that there are many pursuers agents, it is unintuitive why there is a discrepancy between Fixed-k=1 and ART-k=1. Maybe the authors could clarify?_**
>
> Fixed-K=1 uses one fixed adversary, while ART trained with k=1 learns which of the other agents can inflict the maximal damage to the attacked agent when selected as an adversary – something which Fixed-K=1 is oblivious to. Thus, these results demonstrate that it is more advantageous to train against a worst-case adversary.

---

> ### Author Response · Authors · 2024-06-28
>
> **_Question for Section 5.3.Quantitative Analysis: When the authors state "In Independent Spread, optimal performance of ART is achieved if the number of adversaries during training matches their number at deployment time", do they mean "maximum number of adversaries during training"?, since, to the best of my understanding, the number of actual adversaries is determined by $\theta$ and changes across training steps in ART-k._**
>
> Yes, the reviewer is right. It means the maximum number of adversaries sampled during training in each iteration. Due to the natural monotonicity of the loss function with respect to the number of adversaries used, this tends to be the maximum possible.
>
> **_Question for Section 5.3.Convergence: Could the authors clarify what they mean by "A small increase signifies the approximate convergence of ART to an ARNEQ."? First, it is unclear why ART would converge even approximately to an ARNEQ, as this hasn't been shown theoretically to the best of my understanding. Second, could the authors clarify what they mean by "Depicted is the difference between the performance of the improved policy and the original performance." When they say "original performance", do they mean the performance of the originally benign-trained victim policy against the other benign policies or the benign victim policy against the adversarially trained policies? If it's the former, couldn't you argue that even a decrease in performance signifies converges to an ARNEQ?_**
>
> First, as mentioned in previous replies related to convergence, theoretically proving convergence of independent learning to any type of Nash equilibrium in polynomial time is a computationally hard problem. This does not imply that independent learning does not, in fact, converge to an approximate equilibrium, only that proving it is a hard problem. Thus, the point of the plot in Figure 5 is to see whether we achieve any type of stability. We do this by seeing whether further training individual players, while fixing all other learned policies improves the player’s performance any further, which would mean that equilibrium has not been achieved. Figure 5 shows that on average performance tends to stay the same, except for some small oscillations which are due to the stochasticity of deep RL methods. This means that we may have closely approached an equilibrium. Second, what we mean is the performance of the fixed benign agent against other benign agents and its own adversarially trained policies.
>
> **_RC3 for Section 5.3.Convergence: In general, it feels insufficient to say that this experimental setup has anything to do with the ARNEQ. The authors could clarify how this is connected to the ARNEQ more specifically. Otherwise, to strengthen the claim that ART converges to an ARNEQ, it would be much more convincing if you had an environment where the method from Section 3 is feasible to compute and compare ART with the method from Section 3, which would greatly improve the papers results and improve the connection between Section 3 and Section 4. To repeat, I find it sufficient to say "we introduce a model-free gradient-based algorithm that is able to empirically provide an efficient defense in various MARL environments.", which the authors do in Section 3. I find the evidence insufficient to say that "As the achieved improvement is small, this suggests that we approach an ARNEQ.". Even if ART does not approximate ARNEQs, I find the algorithm, empirical evaluation and general motivation interesting for the TMLR audience._**
>
> We have added the phrase “We introduce a model-free gradient-based algorithm that is able to empirically provide an efficient defense in various MARL environments.” at the end of Section 3, as mentioned in a previous response.

---

> > ### Comment · Reviewer_aM2t · 2024-07-02
> > **Acknowledgement of Rebuttal**
> >
> > I thank the authors for the detailed responses, which addressed all my questions and concerns.

---

### Author Response · Authors · 2024-06-28
**To all reviewers**

We thank all the reviewers for their helpful and insightful comments. In the revised version of the paper, we have addressed the reviewers’ concerns. To facilitate readability, we have also reflected these changes in blue in the revised version of the paper.

---

### Decision · Action_Editor_62uh · 2024-07-29

**Recommendation:** Accept as is

**Comment:**

All reviewers agree that the paper is acceptable.

**Audience:**

The paper is interesting to the MARL community.

**Claims And Evidence:**

All reviewers agree that the claims and evidence are sufficient for accepting the paper.